# CLSY docking to Pol IV requires a conserved domain critical for small RNA biogenesis and transposon silencing

Luisa Felgines[1,8], Bart Rymen[1,8], Laura M. Martins[2,8], Guanghui Xu [2], Calvin Matteoli[1], Christophe Himber[1], Ming Zhou[2,3], Josh Eis[2], Ceyda Coruh[2], Marcel Böhrer[1], Lauriane Kuhn [4], Johana Chicher [4], Vijaya Pandey[5], Philippe Hammann [4], James Wohlschlegel [5], Florent Waltz [6], Julie A. Law [2,7] ✉ & Todd Blevins [1] ✉

Eukaryotes must balance the need for gene transcription by RNA polymerase II (Pol II) against the danger of mutations caused by transposable element (TE) proliferation. In plants, these gene expression and TE silencing activities are divided between different RNA polymerases. Specifically, RNA polymerase IV (Pol IV), which evolved from Pol II, transcribes TEs to generate small interfering RNAs (siRNAs) that guide DNA methylation and block TE transcription by Pol II. While the Pol IV complex is recruited to TEs via SNF2-like CLASSY (CLSY) proteins, how Pol IV partners with the CLSYs remains unknown. Here, we identified a conserved CYC-YPMF motif that is specific to Pol IV and is positioned on the complex exterior. Furthermore, we found that this motif is essential for the co-purification of all four CLSYs with Pol IV, but that only one CLSY is present in any given Pol IV complex. These findings support a "one CLSY per Pol IV" model where the CYC-YPMF motif acts as a CLSY-docking site. Indeed, mutations in and around this motif phenocopy *pol iv* null and *clsy* quadruple mutants. Together, these findings provide structural and functional insights into a critical protein feature that distinguishes Pol IV from other RNA polymerases, allowing it to promote genome stability by targeting TEs for silencing.

Eukaryotes maintain genome segments in different chromatin states, including transcriptionally permissive euchromatin and typically silent heterochromatin. Transposable elements (TEs) are frequently marked with chromatin modifications associated with transcriptional silencing, including DNA methylation, which limits potentially dangerous transposition events and protects genome integrity[1]. In animals, this process is directed by PIWI-interacting RNAs (piRNAs), whose precursors are synthesized by RNA polymerase II (Pol II)[2]. By contrast, plants target TEs using specialized transcription machinery, including RNA polymerase IV (Pol IV), to generate small interfering RNAs (siRNAs), and RNA polymerase V (Pol V), to produce nascent long noncoding RNAs, that mediate RNA-directed DNA methylation (RdDM)[3–5].

[1]Institut de Biologie Moléculaire des Plantes, CNRS, Université de Strasbourg, Strasbourg F-67084, France. [2]Plant Molecular and Cellular Biology Laboratory, Salk Institute for Biological Studies, La Jolla, CA 92037, USA. [3]State Key Laboratory of Plant Environmental Resilience, College of Life Sciences, Zhejiang University, Hangzhou 310058, China. [4]Institut de Biologie Moléculaire et Cellulaire, CNRS, Plateforme Protéomique Strasbourg-Esplanade, Strasbourg F-67084, France. [5]Department of Biological Chemistry, University of California, Los Angeles, CA 90095, USA. [6]Biozentrum, University of Basel, CH-4056 Basel, Switzerland. [7]Division of Biological Sciences, University of California, San Diego, La Jolla, CA 92093, USA. [8]These authors contributed equally: Luisa Felgines, Bart Rymen, Laura M. Martins. ✉e-mail: jlaw@salk.edu; todd.blevins@ibmp-cnrs.unistra.fr

Notably, these two plant polymerases do not interact with the key recruitment and initiation factors of Pol II (TFIIs[6]), but instead interact with their own polymerase-specific partners[7–13]. Thus, animal and plant lineages have converged on sequence-specific TE silencing mechanisms via distinct RNA polymerases and small RNA pathways[14–16].

In the model plant Arabidopsis, Pol IV initiates siRNA biogenesis when recruited to chromosomal DNA via four CLASSY (CLSY) proteins (CLSY1/2/3/4)[11,17], which have SNF2 domains similar to ATP-dependent chromatin remodelers[18,19] and specifically interact with Pol IV[10,20] via an unknown mechanism (Fig. 1a). The different CLSYs control siRNA production at distinct subsets of loci throughout the genome[11,21,22]. CLSY1 and CLSY2 function primarily in the chromosome arms, while CLSY3 and CLSY4 act primarily in pericentromeric heterochromatin[11,21]. In addition to this locus-specific targeting, the CLSYs are differentially expressed during plant development[21,23,24] and are required for the tissue-specific regulation of DNA methylation patterns[21,24,25]. In all tissues tested, Pol IV targeting by CLSY1 and CLSY2 requires SAWADEE HOMEODOMAIN HOMOLOG 1 (SHH1), a chromatin reader that binds histone tails with methylated histone 3 lysine 9 (H3K9me) and unmethylated histone 3 lysine 4 (H3K4) residues[7,20]. Mechanistically, CLSY1 and SHH1 directly interact in vitro[20], and CLSY1 and CLSY2, but not CLSY3 and CLSY4, are required to mediate the interaction between SHH1 and the Pol IV complex in vivo[11]. Together, these findings link Pol IV targeting by CLSY1 and CLSY2 to H3K9me modifications. For CLSY3 and CLSY4, the mechanisms for locus-specific targeting remain poorly understood, but they are independent of H3K9 methylation[11], demonstrating distinct protein interactions and modes of Pol IV targeting for CLSY1 and CLSY2 vs. CLSY3 and CLSY4. Additional accessory factors that modify Pol IV targeting and siRNA production at specific genomic sites have also been discovered, including the chromatin reader ZMP[26], demonstrating that Pol IV recruitment is highly regulated.

Once targeted, Pol IV transcription is physically and enzymatically coupled to RNA-DEPENDENT RNA POLYMERASE 2 (RDR2)[10,12,27–29]. Pol IV transcribes DNA into short transcripts[12,30,31] with lengths measured by structural aspects of the Pol IV-RDR2 complex and transcription bubble[29,32]. These short transcripts then serve as templates for RDR2 to synthesize a second RNA strand[29] (Fig. 1a). The resulting ~30 nucleotide (nt) double-stranded RNAs (dsRNAs) are cleaved by DICER-LIKE 3 (DCL3) into 24 nt siRNA duplexes[30,31] (Fig. 1a). These siRNAs are loaded onto ARGONAUTE family proteins (AGO4/6/9), which guide DNA methylation via base-pairing to nascent transcripts synthesized by another plant-specific polymerase, Pol V[33–35].

Both Pol IV and Pol V arose from the duplication and neofunctionalization of genes that encode the 12 subunits of eukaryotic Pol II[3,36–39]. The Pol II, Pol IV, and Pol V complexes are distinguishable by their respective largest subunits, NRPB1, NRPD1, and NRPE1[5,13]. Pol IV-specific functions are thus encoded in its unique largest catalytic subunit, NRPD1 (Fig. 1b), and in its interactions with unique accessory components including RDR2[10,12], SHH1[7,10], and the CLSYs[10,11]. Indeed, cryo-electron microscopy (cryo-EM) of the Arabidopsis Pol IV complex recently revealed that RDR2 and NRPD1 are joined by NRPD1-specific funnel helices (Fig. 1b) that channel RNA templates into the RDR2 active site during Pol IV backtracking[27,40]. However, the features underlying other Pol IV-specific activities remain unclear. For example, it is not known how the CLSYs specifically associate with Pol IV, and it remains unclear why a point mutation in NRPD1, C118Y, that does not affect Pol IV core complex assembly or Pol IV-RDR2 association, has a global effect on siRNA levels[41].

Taken together, previous work has demonstrated that Pol IV evolved from Pol II to couple DNA transcription to dsRNA synthesis in the context of TE-rich chromatin. However, the specific Pol IV features (e.g., amino acids, motifs, etc.) that enable these unique activities have remained elusive. To identify these features, we conducted a phylogenetic analysis of the largest subunits of Pol II (NRPB1), Pol IV (NRPD1)

and Pol V (NRPE1), which revealed exclusively conserved (EC) regions in NRPD1. Of these five EC regions, three (EC3-5) mapped at or near the funnel helices that connect NRPD1 to RDR2 and allow the rapid conversion of Pol IV transcripts into dsRNAs[27,29]. This validates our approach for finding NRPD1-specific features linked to unique Pol IV activities. By leveraging structural, biochemical, and molecular assays, we were then able to map the EC1 and EC2 regions and reveal a key role for EC2 in facilitating Pol IV targeting. Specifically, we found that EC1 and EC2 occupy the previously unresolved Pol IV clamp head region, where four cysteines come together to coordinate a zinc ion: two from EC1 and two from the CYC-YPMF motif[41] identified here as part of EC2. Via a series of immunoprecipitation and mass spectrometry experiments, we found that the CYC-YPMF motif is required for the interaction between Pol IV and the CLSYs and that only one CLSY family member is present in any given Pol IV complex. Overall, these findings support a model in which the CYC-YPMF motif in EC2 acts as a CLSY-docking site. Consistent with these results, mutations in the CYC-YPMF motif, or in another conserved residue in the neighboring clamp core region, resulted in global losses of siRNAs and defects in TE silencing. Together, these structure-function studies identified an exclusively conserved domain within NRPD1 that distinguishes Pol IV from Pol II and Pol V and acts as a CLSY docking site that allows Pol IV to target, transcribe and silence TEs throughout the genome.

## Results
### Identification of five exclusively conserved regions in the largest subunit of Pol IV
To detect protein motifs that represent unique features of Pol IV, we compared the largest subunits of Pol II, Pol IV, and Pol V from diverse plant species. BLASTP queries of the Arabidopsis proteins AtNRPB1, AtNRPD1, and AtNRPE1 against the Phytozome13 and NCBI databases identified numerous close homologs. To pass our quality criteria, we required these proteins to be over 900 amino acids long and contain the catalytic metal A binding site residues (DFDGD) common to all RNA polymerase largest subunits[42–44], resulting in a total of 202 protein sequences from 56 species (Fig. 1c, Supplementary Figs. 1 and 2a; Supplementary Data 1). Phylogenetic analysis based on MUSCLE alignment grouped these subunits of Pol II, Pol IV and Pol V (NRPB1/D1/E1) into four clusters. One clade of 74 protein sequences clustered around AtNRPB1, all of which had heptad repeats in their C-terminal domain (CTD), indicating that they are the largest subunits of Pol II (Fig. 1c, green). As expected given the divergence in the transcription cycles of Pol IV and Pol V compared to Pol II[4,34,45], none of the NRPD1 or NRPE1 proteins have the heptad repeats associated with NRPB1 (Fig. 1c, Supplementary Data 1). Instead, these polymerase subunits are part of two independent clades. AtNRPD1 is in a clade including 58 proteins which contain CYC-YPMF-like motifs (CYC-YPxF) of unknown function[41], as well as the amino acids corresponding to funnel helices that physically connect Pol IV to RDR2 in Arabidopsis[27] (Fig. 1c, purple). AtNRPE1 is in another clade encompassing 58 sequences that contain repeated GW/WG motifs in their CTDs (Fig. 1c, blue). In Arabidopsis, Pol V's GW/WG motifs facilitate the interaction between Pol V and siRNA-loaded AGO proteins to guide de novo DNA methylation[33,46,47].

In contrast to polymerases from the abovementioned vascular plants, the bryophyte NRPD1/NRPE1-like sequences form a distinct, fourth cluster of 12 proteins in our phylogenetic tree (Fig. 1c; teal branches, no background color). One group of proteins in this cluster contains the largest subunit of *Physcomitrium patens* Pol IV (Supplementary Fig. 2a, Ppa_NRPD1)[48]. In three bryophyte species, more than one NRPE1-like protein was detected (Supplementary Fig. 2a). The bryophyte NRPE1-like proteins typically have a shortened CYC-YP motif with divergent amino acids between their 'a' and 'b'-domains, in addition to the expected CTD GW/WG motifs (Supplementary Fig. 2a). However, none of the bryophyte NRPD1 or NRPE1-like proteins had a fully intact CYC-YPxF motif (Supplementary Fig. 2a). In light of the

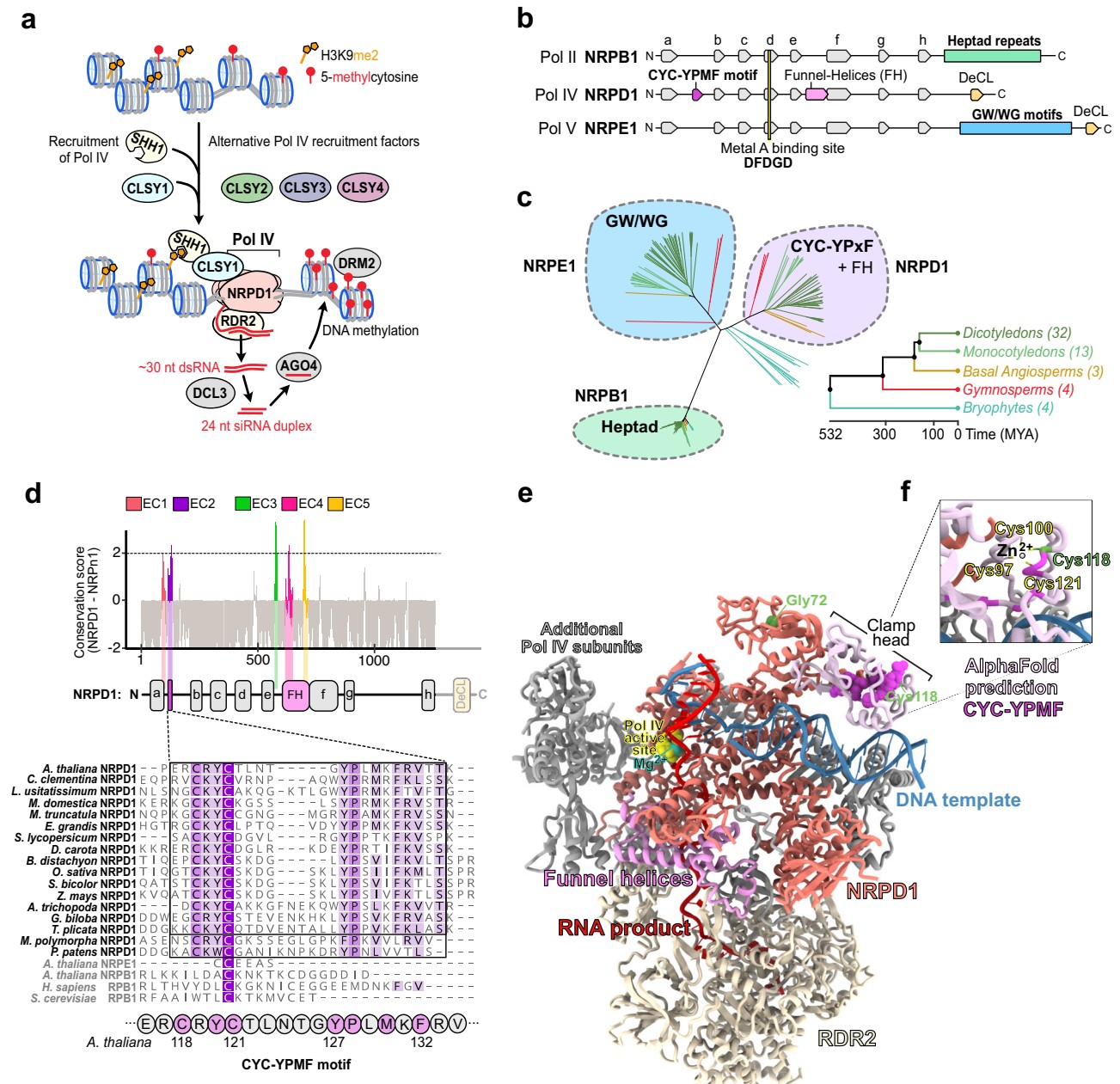

**Fig. 1 | The Pol IV subunit NRPD1 has an exclusively conserved motif exposed at the enzyme exterior. a** Simplified model depicting how Pol IV is recruited to loci for siRNA biogenesis and RdDM. Pol IV recruitment depends on specific partnerships with a chromatin reader SHH1 and four CLASSY proteins (CLSY1/2/3/4) in Arabidopsis. CLSY1 and CLSY2 partner with SHH1, while CLSY3 and CLSY4 function independently of SHH1. Pol IV transcripts are used by RDR2 to synthesize dsRNA that is diced by DCL3 into 24 nt siRNAs that guide DNA methylation. **b** Domain architectures of the largest subunits of Pol II (NRPB1), Pol IV (NRPD1) and Pol V (NRPE1). **c** Phylogenetic analysis of NRPB1, NRPD1 and NRPE1 subunits from 56 species with color shading indicating proteins containing heptad repeats typical of NRPB1 (green), GW/WG motifs typical of NRPE1 (blue) and the CYC-YPxF motif typical of NRPD1 (purple). The species represent basal angiosperms, monocotyledons, dicotyledons, gymnosperms and bryophytes, with the number of species in parentheses and lines colored based on the species groupings (**Supplementary Data 1**). **d** ConSurf analysis of NRPD1 proteins from 52 vascular plant species compared to NRPB1 and NRPE1. Five protein regions that are peaks of NRPD1-exclusive conservation (labeled EC1-EC5) are plotted in color along the

subunit's domain architecture. Positions and domains with low NRPD1 exclusive conservation are in gray. The NRPD1 C-terminal domain, including the DeCL domain, was excluded from this analysis. A multiple alignment of CYC-YPxF motifs from diverse species is shown below with conserved residues in purple and the amino acid numbering based on Arabidopsis NRPD1. **e** A structural model of the Arabidopsis Pol IV-RDR2 (7EU0) complex is displayed as a cartoon representation. NRPD1 is shown mostly in coral, with the Pol IV active site in yellow, a $Mg^{2+}$ bound at the metal A site in teal, the funnel helices in pink, the DNA template in blue and the RNA product of Pol IV in crimson red. The Pol IV clamp head, which was unmodeled in 7EU0, was predicted using AlphaFold2 and is shown in light pink, with its CYC-YPMF motif shown as an atomic sphere representation in bright purple. NRPD1 positions Gly72 and Cys118 (green) are mutated in the *nrpd1-49* and *nrpd1-50* mutants[41], respectively. NRPD2 and most other Pol IV subunits are not shown. **f** A zoomed-in inset details the CYC-YPMF motif position (bright purple). Cys118 and Cys121 of the CYC-YPMF together with Cys97 and Cys100 of the 'a'-domain are predicted to coordinate a zinc ion. This analysis is further detailed in Supplementary Figs. 3, 4.

solid molecular evidence for AtNRPD1's specific role in Arabidopsis Pol IV, ZmNRPD1's specific role in maize Pol IV and OsNRPD1's specific role in rice Pol IV[13,49–51], our phylogenetic analyses suggest that the NRPD1 CYC-YPxF motif arose as a functionally important Pol IV amino acid sequence in a common ancestor of the vascular plants but after their split from the bryophytes.

Having clustered the largest subunits of all three RNA polymerases, we next sought to identify regions that are specifically conserved in NRPD1, as compared to NRPB1 or NRPE1. To this end, we measured the evolutionary conservation[52] of each AtNRPD1 amino acid position relative to corresponding NRPB1 and NRPE1 positions across all the orthologs identified from vascular plants. This analysis identified five major peaks of Pol IV-exclusive conservation (EC1-5), where most amino acids in a nine amino acid window are conserved in NRPD1 but not in NRPB1 or NRPE1 (Fig. 1d). One of these regions, EC1, partly overlaps domain 'a', the first of eight domains that are broadly conserved in the largest subunits of Pol II, Pol IV and Pol V (Fig. 1d, lowercase 'a' to 'h')[4,53], while the others are in regions conserved only in Pol IV (Fig. 1d and Supplementary Fig. 2b). Three of the EC regions (EC3, EC4 and EC5) are between the NRPD1 'e' and 'f'-domains (Fig. 1d). While EC3 is a previously unexamined Pol IV-conserved region, EC4 and EC5 overlap the funnel helices that contribute to RDR2's specific association with Pol IV, rather than with Pol II or Pol V[10,12,27,29]. The other two EC regions neighbor each other in the NRPD1 N-terminus, one overlapping part of the NRPD1 'a'-domain (EC1) and the other covering the CYC-YPxF motif (EC2). While the mechanistic roles of EC1, EC2, and EC3 remain unknown, their strong conservation and prior data showing global reductions in siRNA levels caused by a cysteine to tyrosine mutation in EC2 (YYC-YPMF)[41], suggest that they could impart functions that distinguish Pol IV from Pol II and Pol V.

## The CYC-YPMF motif is part of the NRPD1 clamp head on the exterior of Pol IV

To visualize the three-dimensional (3D) locations of the five EC domains within AtNRPD1, we sought to map them onto the cryo-EM structure of the Arabidopsis Pol IV-RDR2 complex[27] (7EU0). Overall, the AtNRPD1 tertiary structure in 7EU0 is similar to the largest subunits of yeast or mammalian Pol II (RPB1)[27,54–56]. However, the portion of NRPD1 spanning EC1 and EC2, including the CYC-YPMF motif, is unresolved in the 7EU0 structure of the Pol IV-RDR2 complex. Thus, we used an AlphaFold2 prediction of NRPD1 to situate the N-terminal portion of NRPD1 (aa 1-298) within the Pol IV-RDR2 complex using the 7EU0 cryo-EM density data (Fig. 1e and Supplementary Fig. 3). Several lines of evidence support the validity of this positioning. First, the highly conserved 'a' and 'b'-domains that are directly adjacent to the CYC-YPMF motif in AtNRPD1, and form the Pol IV 'clamp core' domain (Supplementary Fig. 3a, green/turquoise residues), are positioned comparably in the 7EU0 structure to those in the AlphaFold2 model of AtNRPD1 (Supplementary Fig. 3a, orange/pink residues). Second, the confidence scores from AlphaFold2 at the CYC-YPMF residues are high (Supplementary Fig. 3b, pLDDT) and the fit of this predicted domain into the experimental Coulomb potential map of Pol IV supports the presence of this globular domain protruding from the clamp core (EMD-31305; Supplementary Fig. 3c). Third, the similarly positioned 'clamp head' domain in Pol II (mammalian 6GMH, yeast 7O75) further reinforces our attribution of this density to the Pol IV region containing the CYC-YPMF. Finally, the electrostatic potential of the putative Pol IV clamp head (Fig. 1e and Supplementary Fig. 3a, purple/pink residues) is positively charged on the surface facing the downstream DNA template, as expected for a domain interacting with DNA (Supplementary Fig. 3d).

Modeling the region of NRPD1 containing EC1 and EC2 within the Pol IV-RDR2 structure (Fig. 1e) allowed visualization of all five EC peaks (Fig. 1d) on the 3D structure of NRPD1, either as a heatmap based on conservation level (Supplementary Fig. 4a) or as discretely colored

regions (Supplementary Fig. 4b). Consistent with their proximity to the NRPD1 funnel helix region, the EC3, EC4 and EC5 regions fold into loops or helices that channel Pol IV RNA transcripts to RDR2. The EC3 loops are adjacent to the primary RNA transcript as it exits Pol IV and enters RDR2, whereas EC4 and EC5 contain conserved NRPD1 amino acids that directly contact RDR2 (N638, E642, Y645, D710, L714)[27] (see Supplementary Fig. 2b), which validates our method for detecting Pol IV-specific features (Supplementary Fig. 4c, bottom inset). Interestingly, we found that two universally conserved amino acids in the RNA polymerase 'a'-domain, Cys97 and Cys100, which are embedded in EC1, combine with Cys118 and Cys121 from EC2 to form the Pol IV clamp head (Fig. 1e and Supplementary Fig. 4d, top inset). Supporting this arrangement, Cys97-Cys100-Cys118-Cys121 likely coordinate a zinc ion together, as detected using Zincbindpredict and AlphaFill (Fig. 1f, zoomed-in inset)[57,58]. The deep conservation of Pol IV-specific Cys118 and Cys121, along with the adjacent YPMF residues, strongly indicates this domain's functional importance to Pol IV. Indeed, an individual C118Y mutation (nrpd1-50) is sufficient to disrupt 24 nt siRNA biogenesis and RdDM[41]. Despite its importance, the function of this motif was not determined[41]. Given that the CYC-YPMF motif is located in the clamp head (Fig. 1e and Supplementary Fig. 3a, purple/light pink domain), which is distant from both the Pol IV active site (Fig. 1e, yellow aspartate triad) and the NRPD1 funnel helices that couple Pol IV to RDR2 (Fig. 1e, central pink residues), it is unlikely to play a direct role in mediating the catalytic activities of Pol IV. Instead, as the CYC-YPMF motif is exposed on the enzyme exterior, we hypothesize that it could facilitate interactions with Pol IV-specific factors that are important for the enzyme's role in RdDM.

## Pol IV association with SHH1 and CLSY recruitment factors requires CYC-YPMF

To test whether the CYC-YPMF motif mediates Pol IV assembly with partner proteins, we transformed nrpd1-3 mutants with constructs encoding one of three variants of the NRPD1 subunit tagged with a 3xFLAG (3xF) epitope: a wild-type variant (NRPD1-3xF$_{WT}$), a variant with a CYC to AAA mutation (NRPD1-3xF$_{AAA-YPMF}$) and a variant with a YPMF to AAAA mutation (NRPD1-3xF$_{CYC-AAAA}$) (Fig. 2a). For each variant, three independent lines expressing similar levels of NRPD1-3xF were selected (Supplementary Fig. 5a) and flower tissue was collected for immunoprecipitation and mass spectrometry (IP-MS) experiments (Supplementary Fig. 5b and Fig. 2b, c, d). In total, six IP-MS datasets were obtained for each NRPD1-3xF variant (i.e., two replicates for each of the three independent lines (Supplementary Data 2)). For all six experiments, the protein spectral counts for each individual component of the purified Pol IV complexes were displayed using balloon plots and colored based on the significance of their enrichment compared to the NRPD1-3xF$_{WT}$ control (Fig. 2b). These plots demonstrate that RDR2, as well as the great majority of Pol IV core subunits (NRPD1 to NRPD12), were observed in all NRPD1-3xF IP-MS replicates (WT, AAA-YPMF or CYC-AAAA; the only subunits not detectable in all replicates were some of the smaller subunits, which could be explained by their digestion into a smaller amount of observable peptides (Fig. 2b)). When the replicate experiments were visualized in aggregate using volcano plots, the Pol IV subunits (green dots) and RDR2 (yellow dot) clustered near the base of the volcano plot (Fig. 2c, d), demonstrating that the Pol IV-RDR2 complex assembles even when the CYC-YPMF motif is mutated. Indeed, of these proteins, the only one passing both the fold change ($|\log_2 FC| \geq 1$) and p-value ($\leq 0.05$) cutoffs was NRPD7b, which is enriched, rather than depleted, in the NRPD1-3xF$_{CYC-AAAA}$ IP-MS experiments (Fig. 2b, d). By contrast, all four CLSY proteins were significantly depleted, and SHH1 was also depleted, though not quite passing the adj. p-value cutoff, in the NRPD1-3xF$_{AAA-YPMF}$ and NRPD1-3xF$_{CYC-AAAA}$ IP-MS experiments compared to the NRPD1-3xF$_{WT}$ control (Fig. 2b, c, d). Indeed, no CLSY or SHH1 peptides were identified in IP-MS experiments using the NRPD1-3xF$_{AAA-YPMF}$ or NRPD1-3xF$_{CYC-AAAA}$ variants.

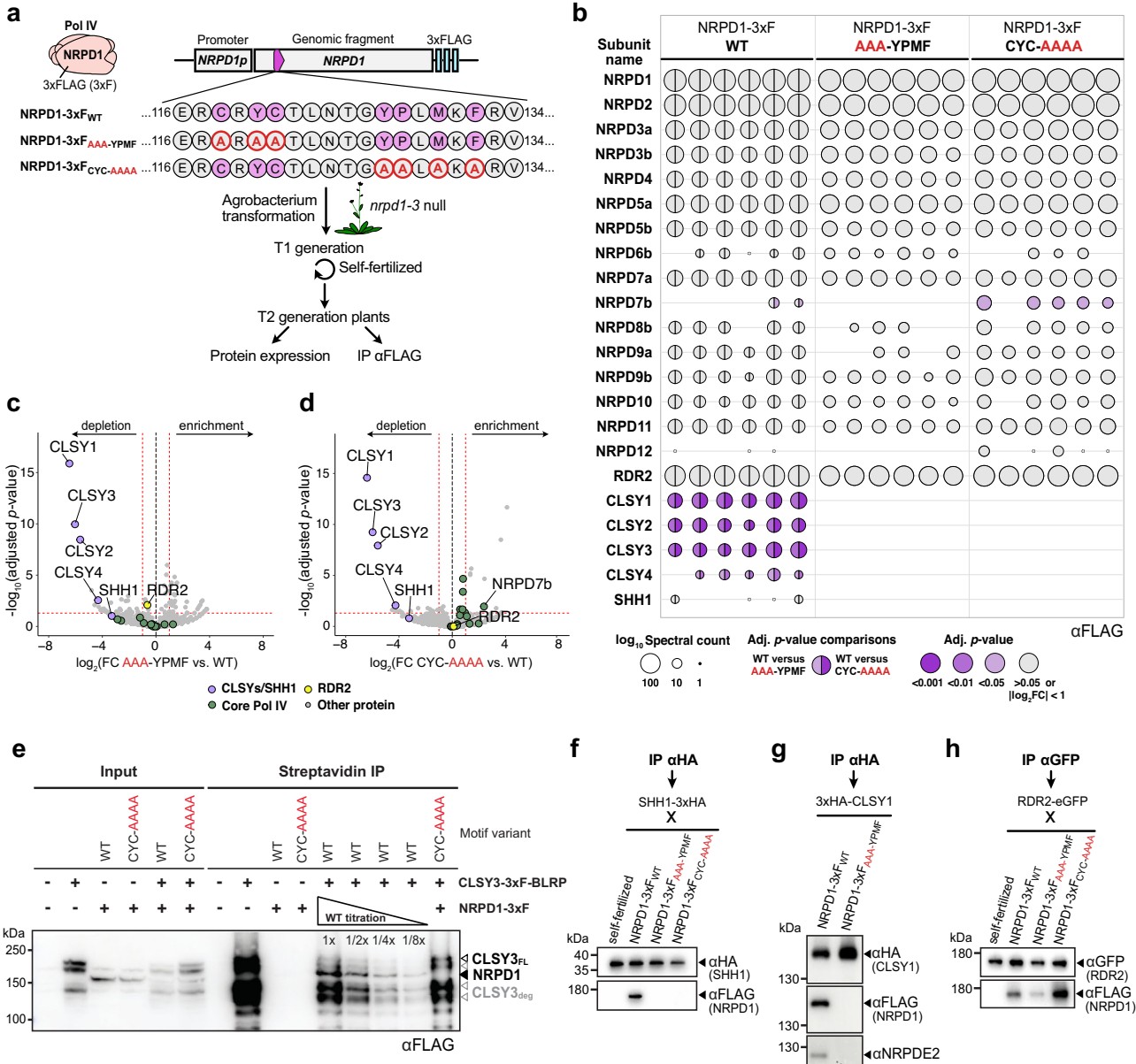

**Fig. 2 | The CYC-YPMF motif is essential for copurification of Pol IV with its recruitment factors. a** Experimental setup to evaluate the effect of mutations in the CYC-YPMF motif on the co-immunoprecipitation of proteins associated with the Pol IV complex (pink) using the NRPD1-3xF variants indicated below. The wild-type AtNRPD1 CYC-YPMF amino acids are indicated in purple and the amino acids altered in the other variant lines are outlined in red. These constructs were transformed into the *nrpd1-3* null mutant and experiments were conducted in the T_2 generation. **b** Balloon plots showing the results from co-immunoprecipitation and mass spectrometry (IP-MS) experiments using flower extracts from three independent lines, each with two technical replicates, expressing the NRPD1-3xF_WT, NRPD1-3xF_AAA-YPMF, or NRPD1-3xF_CYC-AAAA variants. The plot shows protein spectral counts for Pol IV-RDR2 subunits (NRPD1 to NRPD12, RDR2) and other interactors (CLSY1-4 and SHH1) in each dataset. The spectral count is indicated by the $\log_{10}$-transformed balloon area. Purple shaded balloons represent subunits where comparisons between the NRPD1-3xF_WT data and either the NRPD1-3xF_AAA-YPMF or the NRPD1-3xF_CYC-AAAA data pass fold change (FC) and adjusted (adj.) p-value cut-offs, indicating their significance. By contrast, comparisons with |$\log_2$FC| <1 or adj. p-value > 0.05 are shown in gray. In the NRPD1-3xF_WT data, the left half-balloon colors are adj. p-values for comparison to the NRPD1-3xF_AAA-YPMF data, and the right half-balloon colors are adj. p-values for comparison to the NRPD1-3xF_CYC-AAAA data. **c, d** Volcano plots showing the enrichment or depletion of proteins from six

independent IP-MS experiments comparing the NRPD1-3xF_WT lines with either the NRPD1-3xF_AAA-YPMF or the NRPD1-3xF_CYC-AAAA lines. The red hashed lines demarcate a $\log_2$FC of ±1 and an adj. p-value of 0.05. Purple dots indicate Pol IV recruitment factors (CLSY1-4 and SHH1), green dots the Pol IV core subunits, and a yellow dot represents RDR2. The adj. p-values derive from a quasi-likelihood negative binomial generalized log-linear model in IPinquiry4. **e** Anti-FLAG western blot detecting NRPD1 and CLSY3 proteins using the genotypes indicated above each lane from either input or streptavidin IP samples. Unless marked, all samples were loaded at 1x. The bands corresponding to NRPD1 as well as both full length CLSY3 (CLSY3_FL; Black) and several CLSY3 degradation products (CLSY3_deg; Gray) are indicated on the right. **f** Western blots detecting SHH1 and NRPD1 proteins in an Anti-HA IP using flowers from F1 plants of the self-fertilized SHH1-3xHA line, or flowers from this line crossed to the NRPD1-3xF variants indicated. **g** Western blots detecting CLSY1, NRPD1 and NRPD2 in an Anti-HA IP using flowers from 3xHA-CLSY1 plants super-transformed with the NRPD1-3xF variants indicated. **h** Western blots detecting RDR2 and NRPD1 proteins in an Anti-GFP IP using flowers from F1 plants of the self-fertilized RDR2-eGFP line, or flowers from this line crossed to the indicated NRPD1-3xF variants. The western analysis in panel e was performed two times, while the analyses in panels f, g and h were each performed once. Source data are provided as a Source Data file.

Taken together, these data suggest that both the CYC and YPMF sub-motifs of the clamp head are required for Pol IV association with the four CLSY proteins (CLSY1, CLSY2, CLSY3 and CLSY4), as well as with the chromatin reader SHH1.

To confirm the importance of the NRPD1 CYC-YPMF motif in mediating Pol IV's interactions with its recruitment factors via an orthogonal method, smaller scale co-immunoprecipitation (co-IP) experiments were also conducted. For these experiments, transgenes expressing CLSY3-3xF-BLRP[21], 3xHA-CLSY1, SHH1-3xHA, or RDR2-GFP were introduced into plant lines expressing the NRPD1-3xF variants (WT, AAA-YPMF, or CYC-AAAA) via genetic crossing or super-transformation. For CLSY3, the interaction with NRPD1-3xF$_{WT}$ but not with NRPD1-3xF$_{CYC-AAAA}$ was confirmed in several replicate experiments (Fig. 2e and Supplementary Fig. 5c). Moreover, a dilution series of the NRPD1-3xF$_{WT}$ co-IP demonstrated a greater than 8x reduction in the interaction between CLSY3-3xF-BLRP and NRPD1-3xF$_{CYC-AAAA}$ (Fig. 2e). Likewise, the 3xHA-tagged SHH1 and CLSY1 proteins were each found to

co-IP with NRPD1-3xF$_{WT}$ but not with the motif mutant versions of NRPD1-3xF (Fig. 2f, g). These findings, along with the ability of RDR2-GFP to associate with both the WT and mutant NRPD1 variants (Fig. 2h), are consistent with the IP-MS results and further show the critical role of the CYC-YPMF motif in mediating interactions between Pol IV and its targeting factors—the CLSYs and SHH1.

## Pol IV complexes are demarcated by distinct CLSY proteins
Despite recent advances in our understanding of the structure and function of the core Pol IV complex, it remains unknown whether multiple different CLSYs associate with a given Pol IV complex or if each complex contains a dedicated CLSY protein. To address this question, IP-MS experiments were conducted using flower tissue collected from transgenic lines expressing 3xF-tagged versions of all four CLSYs driven by their endogenous promoters and introduced into their respective *clsy* mutant backgrounds to verify functional complementation[21] (Fig. 3a, b, Supplementary Fig. 6 and

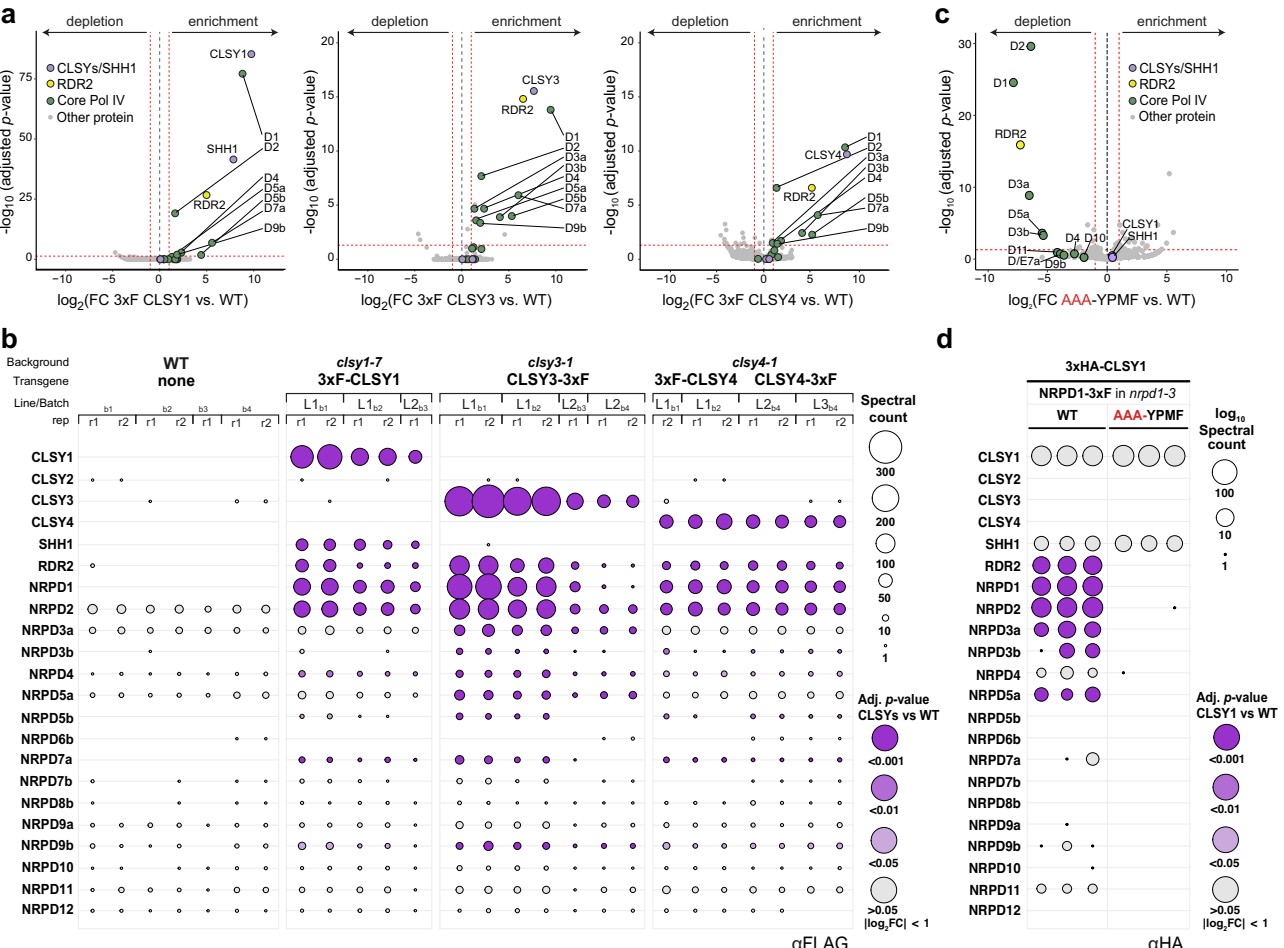

**Fig. 3 | Pol IV complexes only contain one CLSY family member. a** Volcano plots showing the enrichment or depletion of proteins from a combination of several independent CLSY1, CLSY3, and CLSY4 IP-MS experiments compared to non-transgenic control IP-MS experiments. The red hashed lines demarcate a log$_2$FC of ±1 and an adj. *p*-value of 0.05. Enriched Pol IV subunits are designated by subunit number (D1, D2, D3, …, D12) and are colored in green while SHH1 and the CLSYs are colored in light purple and RDR2 in yellow. **b** Balloon plot representation of the individual IP-MS experiments comparing CLSY copurifying proteins to those of non-transgenic WT controls. The spectral counts are indicated by the linear balloon area, and significantly enriched subunits (log$_2$ FC > 1) for the WT vs. 3xF tagged CLSY1, CLSY3, or CLSY4 comparisons are colored based on their adj. *p*-values (gray to purple shades) as detailed in Fig. 2b. Above each plot, the genetic background and transgenes used for each IP are indicated. The line numbers (L#) represent

independent tagged lines, while the batches (b#) represent independent IP-MS experiments, most of which also included technical replicates (r#). **c** Volcano plot as described in panel (**a**), showing the enrichment or depletion of proteins from 3xHA-CLSY1 IP-MS experiments using flowers from the *nrpd1-3* mutant line complemented with either the NRPD1-3xF$_{WT}$ or the NRPD1-3xF$_{AAA-YPMF}$ variant. **d** Balloon plot representation of 3xHA-CLSY1 IP-MS experiments comparing co-immunoprecipitated proteins of 3xHA-CLSY1 in NRPD1-3xF/*nrpd1-3* lines using either the NRPD1-3xF$_{WT}$ or NRPD1-3xF$_{AAA-YPMF}$ variants. The spectral counts are indicated by the log-transformed balloon area, and adj. *p*-values for the comparisons between the NRPD1-3xF$_{WT}$ vs. NRPD1-3xF$_{AAA-YPMF}$ IP-MS experiments are indicated by balloon color as detailed in Fig. 2b. The adj. *p*-values in the volcano and balloon plots derive from a quasi-likelihood negative binomial generalized log-linear model in IPinquiry4.

Supplementary Data 3). Although multiple attempts were made using both N- and C-terminally tagged CLSY2 lines, neither CLSY2 nor Pol IV subunits were detected in IP-MS experiments, suggesting either the abundance of CLSY2 is too low or the epitope is not accessible. However, purifications of the other three CLSYs were robust, as peptides from RDR2 and many Pol IV subunits were significantly enriched across several biological and technical replicates compared to non-transgenic wild-type controls (Fig. 3a, b). Consistent with previous work showing interactions between CLSY1 and SHH1[7,10,11,20], peptides from SHH1 were enriched only in the CLSY1 IP-MS experiments (Fig. 3). Given the high quality and specificity of these CLSY IP-MS experiments, the failure to detect peptides for more than one CLSY (Fig. 3a, b) supports a "one CLSY per Pol IV complex" model.

The aforementioned model, as well as the importance of the CYC-YPMF motif in mediating interactions between Pol IV and its targeting factors, is further supported by an independent set of IP-MS experiments. These experiments utilized flower tissue collected from *nrpd1-3* mutant lines that express either the NRPD1-3xF$_{WT}$ or the NRPD1-3xF$_{AAA-YPMF}$ variant and were super-transformed with a construct encoding the 3xHA-CLSY1 protein (Fig. 3c, d and Supplementary Data 4). 3xHA-CLSY1 and SHH1 peptides were recovered in anti-HA IPs in the presence of either NRPD1-3xF variant (Fig. 3c, d), but no other CLSYs were immunoprecipitated (Fig. 3d), supporting the one CLSY per complex model. Furthermore, the lack of peptides for Pol IV subunits and RDR2 specifically in the IP-MS experiments using the NRPD1-3xF$_{AAA-YPMF}$ variant (Fig. 3c, d) reinforces the importance of the CYC-YPMF motif in mediating the interaction between CLSY1-SHH1 and Pol IV. Taken together, these findings and those presented in Fig. 2, reveal the mechanism that prevents more than one CLSY from associating with Pol IV simultaneously: they all compete for a common NRPD1 docking site comprised of the CYC-YPMF motif. We have thus designated this motif as the "CLSY-docking" motif.

## The CYC-YPMF motif is essential for siRNA production, DNA methylation and TE silencing

To determine the impact of the CYC and YPMF mutations on the activity of the RdDM pathway, siRNA levels were assessed. Specifically, we sought to determine how the severity of these CYC-YPMF mutants compared to each other and to previously identified point mutations at or near this motif within NRPD1[41]. Furthermore, we wondered whether any of these mutants would display selective effects on siRNA clusters controlled by specific CLSY family members. To address these questions, small RNA sequencing (smRNA-seq) experiments were conducted using flower tissue from the three NRPD1-3xF variant lines (WT, AAA-YPMF, or CYC-AAAA) as well as WT, *nrpd1-3*$_{null}$, and *clsy* quadruple (*clsy quad*) mutant controls. These datasets were then compared to a reanalysis of tissue-matched smRNA-seq experiments from two previously identified NRPD1 point mutants (*nrpd1-49*$_{G72E}$ in the clamp core and *nrpd1-50*$_{YYC-YPMF}$ in EC2, the clamp head), along with their respective WT and *nrpd1-51*$_{null}$ controls[41] (Supplementary Data 5). For the aforementioned comparisons, we conducted our analyses using the previously identified 12,939 siRNA clusters, which were categorized based on their *clsy*-dependencies[21]. To confirm the validity of using these clusters, we first demonstrated that they contain >92% of all the 24 nt siRNAs from the control samples in the two datasets being compared (92.9% for SucSul$_{WT}$ and 92.7% for Col-0$_{WT}$; Supplementary Fig. 7a). Furthermore, we demonstrated that ~99% of these clusters were significantly downregulated (log$_2$FC ≤ 1 and FDR < 0.01) in the *clsy* quad and both *nrpd1* null mutants, confirming that these clusters are both CLSY and Pol IV-dependent in the experiments being compared (Supplementary Fig. 7b).

We then determined the relative strengths of the NRPD1-3xF variants and *nrpd1* point mutations across the full set of 12,939 loci via smRNA-seq experiments (Fig. 4a and Supplementary Fig. 7b), or at select loci via northern blotting (Supplementary Fig. 7c and Ferrafiat

et al.[41]), again relying on the vetted siRNA clusters (Supplementary Fig. 7d). While all the mutants showed strong reductions in siRNAs, nearing the severity of *nrpd1* null mutants, some were stronger than others: the NRPD1-3xF$_{CYC-AAAA}$ variant was the strongest, followed closely by the *nrpd1-49*$_{G72E}$ and NRPD1-3xF$_{AAA-YPMF}$ mutants, with the *nrpd1-50*$_{YYC-YPMF}$ mutant being the weakest (Fig. 4a and Supplementary Fig. 7b, e). These data show that the CLSY-docking motif is required for the biogenesis of nearly all Pol IV-dependent siRNAs, with the location of the mutation affecting the severity of the siRNA defects.

Next, we compared the effects of the various *nrpd1* mutants and NRPD1-3xF variants on siRNA levels at clusters dependent on different combinations of CLSY proteins. As expected, the two *nrpd1* null mutants and the *clsy quad* mutant have the strongest effects across all categories (Fig. 4b). For the NRPD1-3xF$_{CYC-AAAA}$, NRPD1-3xF$_{AAA-YPMF}$, and *nrpd1-50*$_{YYC-YPMF}$ mutants, the siRNA defects were also similar across all categories: NRPD1-3xF$_{CYC-AAAA}$ was always the strongest, NRPD1-3xF$_{AAA-YPMF}$ was always intermediate, and *nrpd1-50*$_{YYC-YPMF}$ was always the weakest (Fig. 4b). Thus, these mutations do not appear to selectively affect loci regulated by specific CLSY proteins. However, in the *nrpd1-49*$_{G72E}$ mutant, the *clsy3,4*-dependent clusters were significantly less reduced as compared to the other categories (Fig. 4b). In fact, at these clusters, the *nrpd1-49*$_{G72E}$ and *nrpd1-50*$_{YYC-YPMF}$ mutants had similar effects (Fig. 4b, red arrow). Assessment of the behavior at *clsy3*- and *clsy4*-dependent siRNA clusters revealed that most of this difference is due to a weaker effect of the *nrpd1-49*$_{G72E}$ mutant on *clsy3*-dependent clusters (Fig. 4b, red arrow). These same trends in terms of the overall strengths for each mutant and the selectively weaker effect of the *nrpd1-49*$_{G72E}$ mutant on *clsy3*-dependent clusters are also evident when assessing the normalized read values for each replicate sample across all the 24 nt siRNA categories (Supplementary Fig. 7e). Given the clustering of these mutations in the clamp core and head of NRPD1 (Fig. 4b, model inset), which includes the CLSY-docking motif (CYC-YPMF, Supplementary Fig. 3a), these mutant forms of Pol IV are likely all impaired in their association with CLSYs. However, the selective behavior of the *nrpd1-49*$_{G72E}$ mutant on certain Pol IV targets suggests that the location and/or type of *nrpd1* mutation may affect the association of some CLSYs more than others.

To determine the downstream consequences of mutations in the Pol IV clamp head on DNA methylation and TE silencing, the behaviors of the NRPD1-3xF$_{AAA-YPMF}$ and NRPD1-3xF$_{CYC-AAAA}$ variants were compared to WT, *nrpd1-3* null, and *clsy* quad mutant controls. Quantitative Chop-PCR assays, described previously[41], detected DNA methylation at the *AtSN1* retroelement that is lost in *nrpd1-3* null mutants and rescued after transformation with the NRPD1-3xF$_{WT}$ variant (Fig. 4c). Neither the NRPD1-3xF$_{AAA-YPMF}$ nor the NRPD1-3xF$_{CYC-AAAA}$ variant rescued *AtSN1* methylation, as these lines showed methylation levels similar to the *clsy* quad mutant (Fig. 4c). In accord with our Chop-PCR data, RT-qPCR experiments detected *AtSN1* transcripts at higher levels in *nrpd1-3* mutants (where DNA methylation is lost) than in WT plants, and *AtSN1* transcripts were reduced in NRPD1-3xF$_{WT}$ lines due to the functional rescue of *nrpd1-3* permitting *AtSN1* silencing. Finally, the higher accumulation of *AtSN1* transcripts in the NRPD1-3xF$_{AAA-YPMF}$ and NRPD1-3xF$_{CYC-AAAA}$ variant lines (Fig. 4d) demonstrates that the CLSY-docking motif is required for the silencing of *AtSN1* retroelements.

Previous studies showed that Pol IV plays an important role in silencing LTR/*Copia* retrotransposons of the *ONSEN* family, which are activated by 37 °C heat stress[16,41,59,60]. Thus, to assess the dependence of *ONSEN* silencing on the CYC-YPMF motif, we utilized the three NRPD1-3xF variant lines. *ONSEN* transcripts are nearly undetectable by RT-qPCR in Arabidopsis grown under 21/18 °C (day/night) conditions (Fig. 4e, blue panel). After 24 h of 37 °C treatment *ONSEN* transcripts accumulate to a much higher level in *nrpd1-3* null plants compared to WT (Col-0$_{WT}$) controls (Fig. 4e, orange panel). Moreover, *ONSEN* levels are at WT levels in the NRPD1-3xF$_{WT}$ variant lines but are higher in the NRPD1-3xF$_{AAA-YPMF}$ and NRPD1-3xF$_{CYC-AAAA}$ variant lines, with NRPD1-

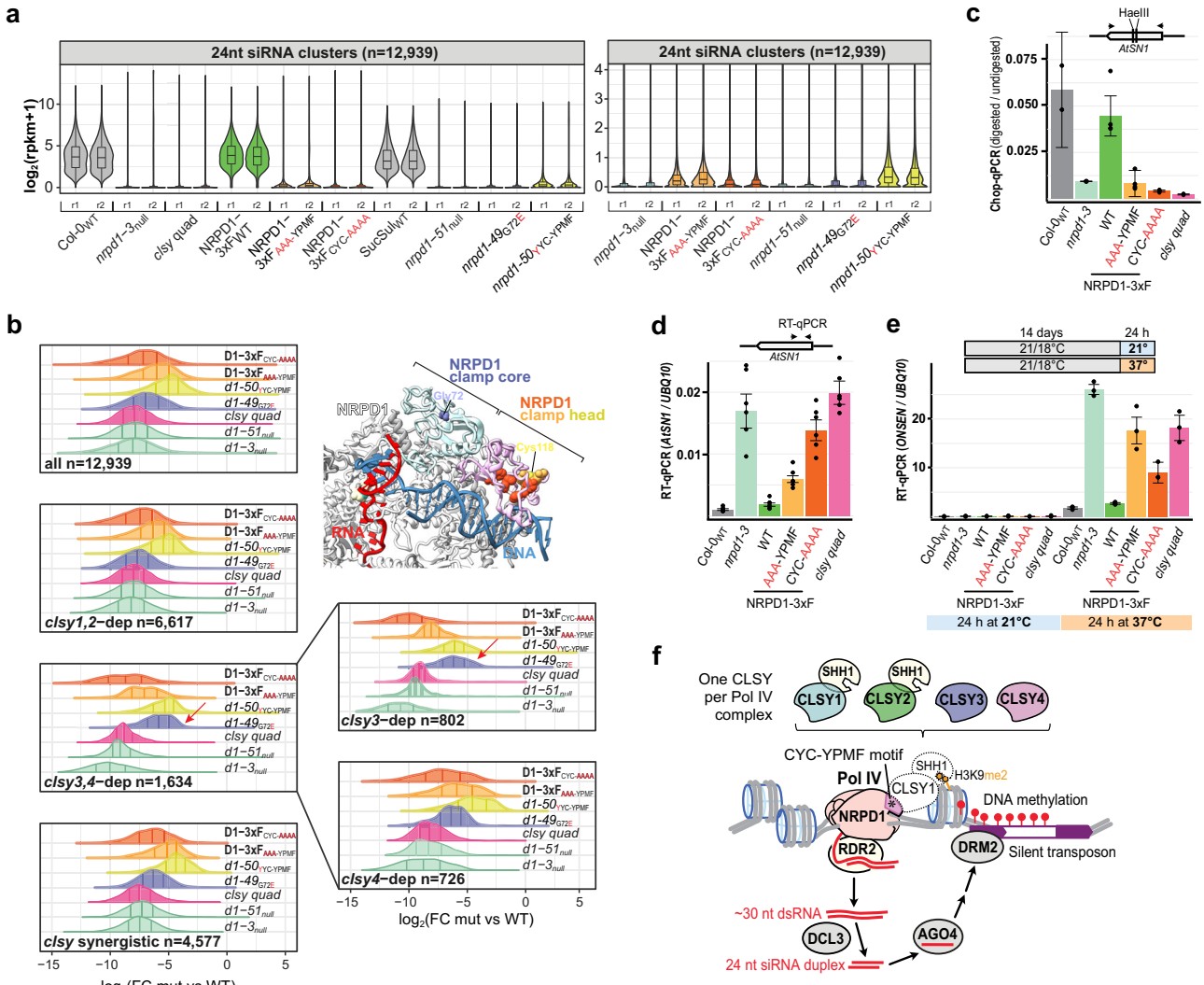

**Fig. 4 | The CLSY-NRPD1 interaction is important for siRNA biogenesis and TE silencing. a** Violin plots showing the normalized expression levels [$\log_2$(rpkm+1)] of 24 nt siRNAs across the full set of clusters ($n = 12{,}939$) in the genotypes indicated below for two biological replicates (r1 and r2). Select genotypes are shown on a smaller Y-axis scale on the right. Each boxplot displays the median, first and third quartiles (lower and upper hinges), the largest value within 1.5 times the interquartile range above the upper hinge (upper whisker), and the smallest value within 1.5 times the interquartile range below the lower hinge (lower whiskers). **b** Half violin plots showing the differential expression profiles ($\log_2$FC) of siRNA clusters in the indicated categories (lower left) for each mutant (right) relative to their respective wild-type controls. The NRPD1 and *nrpd1* symbols are abbreviated as D1 and *d1*, respectively. Red arrows highlight the siRNA expression profiles that are shifted relative to the rest of the categories for the *nrpd1-49*$_{G72E}$ mutant. A portion of the Pol IV structure showing the locations of the characterized mutants is shown as an inlay. **c** Methylation at the *AtSN1* retroelement evaluated by Chop-qPCR. Genomic DNA was digested with a methylation-sensitive restriction enzyme (HaeIII). DNA methylation levels in each sample, which protect HaeIII sites in *AtSN1*

from cleavage, were detected by qPCR with primers flanking the sites and normalized using undigested controls. RT-qPCR to quantify the transcript level of (**d**) *AtSN1* under optimum growth conditions and (**e**) *ONSEN* expression in response to 24 h 37 °C heat or without treatment (mock). Relative expression in the genotypes indicated below were normalized to the *UBQ10* transcript level as a reference for both *AtSN1* and *ONSEN*. The *ONSEN* assays were further normalized to the Col-0$_{WT}$ mock level. For all qPCR analyses (**c**–**e**), the error bars represent standard deviation from the mean of technical replicates (In the RT-qPCR assays, $n = 6$ for *AtSN1* and $n = 3$ for *ONSEN*. In the Chop-qPCR analysis, $n = 2$ for the Col-0, *nrpd1* and *clsy quad* samples, and $n = 4$ for the NRPD1-3xF lines). **f** Model for CLSY-Pol IV docking via the CYC-YPMF motif in the NRPD1 clamp head (asterisk) where there is only one CLSY family member per Pol IV complex. While the locus-specific targeting of RdDM relies on different combinations of CLSY-Pol IV complexes depending on the tissue, as an example, here we show CLSY1 initiating 24 nt siRNA biogenesis and RdDM. At CLSY1-dependent sites Pol IV is recruited using the SHH1 protein, which forms a CLSY1-SHH1 subcomplex that reads H3K9 dimethylation via the Sawadee domain of SHH1. Source data are provided as a Source Data file.

$3xF_{AAA\text{-}YPMF}$ plants showing *ONSEN* levels that phenocopy *clsy* quad mutants (Fig. 4e). Together, these findings demonstrate the importance of the CLSY-Pol IV docking motif residues (i.e., CYC-YPMF) for RdDM, for the silencing of *AtSN1* loci, and for the repression of heat-activated *ONSEN* retrotransposons.

## Discussion

Although RNA Pol IV evolved from Pol II, these two polymerases serve very distinct functions: Pol II transcribes coding regions into mRNAs,

while Pol IV works with RDR2 to transcribe TEs and repeats into short dsRNAs that ultimately target DNA methylation and facilitate gene silencing. One critical distinction between these RNA polymerases is how they are recruited to chromatin. Pol II is targeted to defined sequence elements via associations with both specific and general transcription factors[6], while Pol IV is targeted to distinct chromatin environments via associations with the CLSY family of SNF2-like proteins[17]. Understanding how the structurally-similar Pol II and Pol IV complexes are directed to different targets is critical because the

mistargeting of essential genes for silencing by Pol IV or the mistargeting of TEs for expression by Pol II are both dangerous off-target effects. Findings that the CLSYs and RDR2 associate with Pol IV, but with none of the other RNA polymerases[10–13,61,62], provided the first insights into the unique ability of Pol IV to generate dsRNA. Here we uncovered the mechanism enabling the CLSY proteins to specifically associate with Pol IV and demonstrated this mechanism's importance in regulating the epigenome.

Specifically, we discovered that NRPD1, a Pol IV-specific subunit, contains a conserved CYC-YPMF motif that is critical for mediating the associations of all four CLSYs with the Pol IV complex and for regulating DNA methylation and siRNA production throughout the genome. In addition, we found that only one CLSY family member associates with a given Pol IV complex. These findings support a "one CLSY per Pol IV" model in which the CLSYs compete for binding to the CYC-YPMF "CLSY-docking" motif to control the distribution of Pol IV across the genome (Fig. 4f). Taken together, our work has uncovered a structural innovation within NRPD1 that serves dual roles in controlling DNA methylation patterns. First, it ensures that the CLSYs only interact with Pol IV and second, it creates a highly tunable layer of regulation for Pol IV targeting based on the abundances of the CLSYs in each cell. More generally, the separate docking of four CLSYs with Pol IV via a common motif represents a previously unknown mechanism by which SNF2 proteins can, as a group, coordinate noncoding transcription and small RNA biogenesis to maintain genome stability in eukaryotes.

Overall, the identification and investigation of amino acid regions that are exclusively conserved in the largest subunit of Pol IV (as compared to Pol II and Pol V) has provided several major insights into how Pol IV carries out its distinct roles in producing dsRNAs and mediating gene silencing. By applying a probabilistic evolutionary model[63] to a phylogenetic analysis of ~190 RNA polymerases (Fig. 1c, Supplementary Fig. 1 and Supplementary Data 1), we identified five regions that are exclusively conserved in NRPD1 (EC1-5; Supplementary Table 3). One of these regions, EC2, contains the CYC-YPMF motif that we have now demonstrated acts as a CLSY-docking motif. This fulfills the critical function of ensuring the targeting of Pol IV, but not Pol II, to TEs and repeats throughout the genome. In the recently published cryo-EM structure of the Pol IV-RDR2 complex[27], two of the other EC regions, EC4 and EC5, were shown to form funnel helices that mediate interactions between RDR2 and NRPD1. This Pol IV-RDR2 structural work, the structural analysis of recombinant RDR2[40] and detailed in vitro studies of Pol IV and RDR2 activities[12,28,29,40] all support a model in which stalling of Pol IV transcription initiates backtracking of the Pol IV complex on the DNA template and threading of its RNA product to the active site of RDR2 via an interpolymerase channel. Thus, as compared to EC2, the EC4 and EC5 regions fulfill a separate, but equally critical, function by ensuring that the synthesis of dsRNA by RDR2 is coupled to transcription by Pol IV, but not by Pol II or Pol V. For the remaining two EC regions, EC1 and EC3, functional data is currently lacking. However, their conservation and positioning within the Pol IV complex suggest testable hypotheses about roles they might play to mediate Pol IV-specific functions. As discussed more below, EC1 is intimately connected with EC2 and thus may also contribute to CLSY binding. For EC3, the position of this loop within the channel connecting Pol IV and RDR2 suggests it could play an important role in guiding the Pol IV transcript into the RDR2 active site.

Determining the locations of conserved regions within the Pol IV-RDR2 structure has also proved informative in assessing their contributions to Pol IV function. By combining an AlphaFold2 prediction of NRPD1 with the Pol IV-RDR2 cryo-EM structure[27], our resolution of the NRPD1 region corresponding to EC1 and EC2 yielded a series of key insights. First, we found that residues from EC1 and EC2 come together to coordinate a zinc ion within the clamp head region of Pol IV (Fig. 1f and Supplementary Fig. 4d). Second, we observed that the CYC-YPMF motif of EC2 is exposed at the enzyme exterior on one side of the

clamp head (Fig. 1e), consistent with its role in mediating Pol IV-specific interactions with the CLSYs. Third, we demonstrated that the other side of the clamp head is lined with positively charged residues predicted to make contact with the downstream DNA template (Supplementary Fig. 3d). Finally, though Pol IV and Pol V fail to copurify with general transcription factors like TFIIA, B, D, E, F or H[10,13,27,64], comparison of the Pol IV clamp head to the homologous Pol II domain suggests possible activities CLSY proteins could have during Pol IV transcription initiation. The clamp head in Pol II (RPB1 subunit) contacts a Pol II-specific transcription factor TFIIE, which recruits the TFIIH translocase that initially opens downstream DNA[6,56]. By analogy, a potential mechanism for CLSY function in Pol IV transcription could be to bind the chromatin of transposons, docked to Pol IV via its clamp head, and unwind the downstream DNA. As the timing of the associations between the CLSYs and Pol IV remains unclear, individual CLSYs could bind DNA without Pol IV, followed by Pol IV docking to unwind and transcribe DNA, or they could bind and open DNA prior to Pol IV docking, or the Pol IV complex, already docked with one of the CLSYs, could target and transcribe DNA together as coordinated yet distinct biochemical activities. In Fig. 4f, we present one plausible version of this process, in which CLSY1-SHH1 recruitment factors facilitate Pol IV transcription into downstream DNA at H3K9me2-marked loci by recruiting Pol IV via its CLSY docking motif.

Notably, the ability of the CYC-YPMF motif to facilitate the association of any one of the four CLSYs with the Pol IV complex distinguishes it from other known Pol IV-specific interactions, which has important consequences for the regulation of DNA methylation patterns. Although RDR2 and SHH1 each have close paralogs in Arabidopsis, only these specific family members associate with the Pol IV complex, demonstrating a high degree of specificity in their binding properties. By comparison, the CYC-YPMF docking site in NRPD1 is more versatile, as it can mediate interactions with all four CLSY partners, suggesting a scenario where the CLSYs compete for binding to Pol IV (Fig. 4f). While the rules governing such competition remain to be elucidated, presuming some CLSYs are preferred over others, it would add another dimension to their regulation of DNA methylation. Past studies have already demonstrated that the CLSYs are differentially expressed during development and that they target Pol IV to distinct genomic targets[11,21], resulting in tissue-specific methylation patterns and links between specific CLSYs and several epigenetically regulated traits[17,65]. Within this context, our discovery of a CLSY docking motif in Pol IV suggests a new layer of regulation that could control DNA methylation patterns: competition for binding to the Pol IV complex. Such competition would allow the relative fractions of Pol IV complexes associated with each CLSY to be contextually modulated and also readily reprogrammable. For example, the genomic positions and amounts of DNA methylation could change based not just on the level of each specific CLSY but also on which family members are co-expressed (e.g., Equivalent amounts of CLSY1 expressed with a CLSY that competes better for association with Pol IV versus one that competes worse would lead to different pools of Pol IV complexes, with more Pol IV-CLSY1 complexes in the latter versus the former scenario). Along these same lines, dramatic increases in the expression of a single *CLSY* gene could rapidly redirect Pol IV by driving the equilibrium of Pol IV composition towards complexes limited to a single CLSY family member. Finally, unlike the mutations in the CYC-YPMF motif that affect siRNA levels similarly regardless of the CLSY responsible, the G72E mutation has a smaller effect on siRNA levels at loci controlled by CLSY3, hinting at another possible layer of regulation via natural variation within the NRPD1 subunit of Pol IV. While the aforementioned modes of regulation by the CLSYs remain to be tested, different combinations of these regulatory strategies could explain how a family of just four factors can span the full gamut of DNA methylation regulation—from tuning methylation levels in response to the environment[66] to generating epigenetic diversity

during development[21,23,67] to germline reprograming during sexual reproduction[24,25].

In summary, our findings uncovered a genetic innovation within the Pol IV clamp head (the CYC-YPMF motif) that distinguishes it from other RNA polymerases and enables its targeting to TEs and repeats by mediating interactions with the CLSY proteins. This newly discovered docking location for CLSYs, along with their previously demonstrated targeting functions and the future characterization of their putative chromatin remodeling activities, helps explain how Pol IV functions without the aid of TFII general transcription factors that are critical for Pol II recruitment and initiation. To further test our model, CLSY motifs that facilitate docking to Pol IV and differentiate their activities from other RNA polymerase recruitment factors could be explored by comparing amino acids in the CLSYs to those of DRD1, an SNF2 domain protein that is required for Pol V transcription. Intriguingly, CLSY functional diversity appears to have increased in flowering plants, which typically have three or more distinct CLSYs, whereas non-vascular plants such as *P. patens* have only a single, CLSY1-like protein[37,68]. Given the existence of other specialized factors that enable RNA polymerases to transcribe in heterochromatin, like the TFIIA-related Moonshiner protein in Drosophila that directs Pol II to initiate piRNA production[14], our evolutionary rate and structural modeling approaches could provide further insights into atypical Pol II recruitment factors, or more broadly, into protein families with specialized activities in other processes and species.

## Methods

### Plant materials
**Genetic mutants.** Previously published Arabidopsis T-DNA insertion mutants used in this study include: *nrpd1-3* (SALK_128428)[69], *nrpd1-4* (SALK_083051)[70], *clsy1-7* (SALK_018319)[71], *clsy2-1* (GABI-Kat line 554E02), *clsy3-1* (SALK_040366) and *clsy4-1* (SALK_003876)[11]. The *nrpd1-49*, *nrpd1-50* and *nrpd1-51* point mutations were described by Ferrafiat et al.[41].

**NRPD1 epitope tagged lines.** The *nrpd1-3* null mutant was transformed using Agrobacterium with different *NRPD1p::NRPD1-3xF* (BastaR) constructs that express either WT (NRPD1-3xF$_{WT}$) or mutant (NRPD1-3xF$_{AAA-YPMF}$ or NRPD1-3xF$_{CYC-AAAA}$) forms of the NRPD1-3xF protein. Basta resistant T$_2$ progeny homozygous for the *nrpd1-3* mutation in which the NRPD1-3xF protein was strongly expressed (Supplementary Fig. 5a) were selected for IP-MS experiments. In separate experiments, these lines were super-transformed with a *CLSY1p::3xHA-CLSY1* (HygR) construct and the resulting T$_2$ progeny were selected for co-IP and IP-MS experiments based on hygromycin resistance as well as the accumulation of the 3xHA-CLSY1 and NRPD1-3xF proteins.

**RDR2, SHH1 and CLSY epitope tagged lines.** Arabidopsis Col-0 plants were transformed using Agrobacterium with *RDR2p::RDR2-eGFP* or *SHH1p::SHH1-3xHA* constructs. T$_2$ progeny were selected based on hygromycin resistance and detection of the RDR2-eGFP or SHH1-3xHA proteins. T$_2$ lines were crossed with the NRPD1-3xF$_{WT}$, NRPD1-3xF$_{AAA-YPMF}$, or NRPD1-3xF$_{CYC-AAAA}$ lines and F$_1$ progeny were used for co-IP experiments.

All the N- and C-terminal 3xF tagged CLSY1, CLSY3 and CLSY4 lines used here are in their respective *clsy* mutant backgrounds and are driven by their respective endogenous promoters. Four lines were previously characterized and shown to complement their respective *clsy* mutant phenotypes by Zhou et al.[21] and the remaining lines were characterized in this study (Supplementary Fig. 6). In all cases, T$_3$ lines homozygous for the various 3xF-tagged CLSYs were identified based on drug selection using Hygromycin and seeds from subsequent generations were grown directly on soil under Salk greenhouse conditions for the IP-MS experiments. For the co-IPs between CLSY3 and either the NRPD1-3xF$_{WT}$ or NRPD1-3xF$_{CYC-AAAA}$ lines, a different CLSY3

tagged line was used that contains both a 3xF epitope and a Biotin Ligase Recognition Peptide (CLSY3-3xF-BLRP) that once biotinylated can be captured using streptavidin beads. This construct, which is also driven by the endogenous *CLSY3* promoter and complements the *clsy3* mutation[21], was crossed to the NRPD1-3xF$_{WT}$ or NRPD1-3xF$_{CYC-AAAA}$ lines and F$_1$ progeny that were double drug selected (Hygromycin and Basta) were used for the co-IP experiments (Fig. 2e and Supplementary Fig. 5c).

### Antibodies
The largest subunit of Arabidopsis Pol IV was detected using a native antibody against a peptide from the AtNRPD1 C-terminus. The second largest subunit of Pol IV was detected using an antibody against a peptide from the AtNRPD2 N-terminus (see Ferrafiat et al.[41]). The various 3xF tagged NRPD1 and CLSY proteins were detected using a monoclonal anti-FLAG-HRP antibody (Sigma #A8592). RDR2-eGFP was detected using an anti-GFP polyclonal antibody[72], whereas 3xHA-CLSY1 and SHH1-3xHA were detected using a monoclonal anti-HA-HRP antibody (Sigma #H6533).

### Molecular cloning and complementation assays
Cloning of the N- and C-terminal 3xF and 3xF-BLRP CLSY lines are as previously described in Zhou et al.[21]. For all the other Arabidopsis transgenic lines, the plant transformation vectors used were generated via the MultiSite Gateway approach (Thermo Scientific). Genomic sequences and promoters were PCR amplified from WT Col-0 genomic DNA using primers flanked with appropriate attB sequences on their 5'-ends. The amplified WT gene sequence was cloned into the pDONR221 vector and its promoter was cloned into pDONRP4p1r using BP Clonase II (Thermo Scientific). To obtain NRPD1 motif mutant lines, Gibson Assembly (NEB) using primers for site-directed mutagenesis was performed on the WT *NRPD1* genomic fragment previously cloned into pDONR221 via the BP Clonase II reaction (pENTR-NRPD1_WT). Then, an LR Clonase II reaction was performed to assemble each *NRPD1p::NRPD1-3xF* sequence in the pB7m34GW plant expression vector (BastaR), or to assemble the *SHH1p::SHH1-3xHA*, *RDR2p::RDR2-eGFP* or *CLSY1p::3xHA-CLSY1* sequences in the pH7m34GW vector (HygR). The expression vectors were validated by sequencing before plant transformation using *Agrobacterium tumefaciens* GV3101.

For the 3xF tagged CLSY lines used in Fig. 3a, b, complementation was assessed by genome-wide smRNA-seq experiments and visualized as volcano plots as described in Zhou et al.[21]. In addition to the lines published in Zhou et al.[21], which correspond to CLSY1 Line1 (ins#1 in Zhou et al.[21]), CLSY3 Line 2 (ins#3 in Zhou et al.[21]), and CLSY4 Line 2 (ins#1 in Zhou et al.[21]), the CLSY1 Line 2, CLSY3 Line 1, and CLSY4 Line 1 materials were sequenced and processed in parallel and thus can be compared to the previously published controls as shown in Supplementary Fig. 6a-c. The coverage, mapping, and size distributions of these new smRNA-seq samples are included in Supplementary Data 5.

### Phylogenetic and exclusive conservation analyses
The amino acid sequences from the largest subunits of the three DNA-dependent RNA polymerases (Pol II, Pol IV, and Pol V) were obtained from Phytozome (https://phytozome-next.jgi.doe.gov/) and the National Center for Biotechnology Information (NCBI) using BLASTP with the queries AtNRPB1, AtNRPD1 or AtNRPE1, respectively. In total, 202 sequences from 56 species were analyzed in Geneious (v11.1.5, https://www.geneious.com/). All multiple sequence alignments were performed using MUSCLE (v3.8.425, default parameters). Phylogenetic trees were built with Geneious Tree Builder using default parameters. Conservation scores were calculated using the ConSurf analysis tool, with AtNRPD1 as the reference sequence and the default settings[52]. The positional conservation in AtNRPD1 was calculated via comparison to all NRPB1, NRPD1, or NRPE1 subunits using an alignment of the plant

sequences recognizable as Pol II, Pol IV, or Pol V largest subunits (Supplementary Data 1) but removing the 12 bryophyte proteins in NRPD1/E1-like subclades (final dataset: 190 proteins from 52 species). To quantify exclusive conservation (EC) at individual AtNRPD1 sites, we took the NRPD1 conservation score and subtracted the maximum of NRPB1 and NRPE1 conservation scores at each position. A high $EC_{NRPD1}$ score represents an amino acid specific to NRPD1, whereas a low $EC_{NRPD1}$ indicates that the amino acid there is similar to at least NRPB1 or NRPE1 (or specific to one of these). In order to identify exclusively conserved regions (EC1, EC2, EC3 …), a centered moving average (k = 9) was applied to $EC_{NRPD1}$. Positions with averages ≥2 were found, then all positive consecutive values surrounding these seed positions were selected and expanded to include positions with positive values separated by <9 aa. These $EC_{NRPD1}$ scores and hotspots were then plotted using R (Fig. 1d). C-terminal domains (CTDs) of the NRPB1, NRPD1 and NRPE1 subunits were masked for the ConSurf plot, because NRPB1 and NRPE1 have highly repetitive and divergent CTDs that do not show the consistent alignment of homologous positions for all species.

## Structural analysis

All structural analyses were performed in ChimeraX v1.3 to v1.6[73]. Inspection of the cryo-EM density from AtNRPD1 in Pol IV-RDR2 (EMD-31305) after Gaussian filtering revealed the presence of a globular domain corresponding to an unmodeled portion (aa 93-203) of the atomic model 7EU0. The structure of the unmodeled Pol IV domain was predicted de novo by selecting the N-terminal region of AtNRPD1 (aa 1-298 from UniProt: Q9LQ02) and running AlphaFold2 on this sequence to obtain the top-ranked prediction via the "monomer_ptm" preset[74]. This AlphaFold2 model was then aligned with the AtNRPD1 model 7EU0 using Matchmaker[75]. The CYC-YPMF domain of interest in the predicted model was then rigid-body fitted to the experimental cryo-EM density. The pLDDT score combined with the proper fit of the domain in the density and the presence of a similarly positioned domain in Pol II structures of *Sus scrofa* (6GMH) and yeast (7O75) confirmed the proper attribution of the density to this unmodeled portion of AtNRPD1 (Supplementary Fig. 3a–c). The AlphaFold2-predicted structure (Fig. 1e, pink residues) overlaps amino acids that have been previously assigned to a putative Pol IV 'clamp head' (aa 85-219)[27], again based on comparison to Pol II structures.

The AtNRPD1 residues Cys97, Cys100, Cys118 and Cys121 were identified as a putative zinc binding site using the online Zincbind-predict tool[57] and supported by the AlphaFill software also predicting a zinc ion at this site[58]. As a positive control, another zinc binding site in AtNRPD1, composed of residues Cys56, Cys59, Cys67, and His70, was identified by Zincbindpredict. These residues are conserved in mammals and yeast and are known to coordinate zinc[53]. For Supplementary Fig. 4a, the atomic model was colored according to the ConSurf analysis using the command "color byattribute" and an attribute file containing the centered moving average of NRPD1 exclusive conservation ($EC_{NRPD1}$) values. The NRPD1 model in Supplementary Fig. 4a displays this moving average as a continuous color gradient ranging from ≥0 (white) to 3.5 (purple). Discrete colors were assigned in Supplementary Fig. 4b to the five NRPD1 regions with $EC_{NRPD1}$ peaks (EC1-EC5) shown in Fig. 1d. The electrostatic potential analysis was performed using the "coulombic" command[73].

## Denaturing protein extraction and western blot analyses

Samples for steady-state protein analyses (Supplementary Fig. 5a) were obtained using a phenol-based extraction on frozen powder from Arabidopsis flowers[76]. The resulting pellets were resuspended in resuspension buffer (RB; Supplementary Table 2) and proteins levels were quantified by Lowry (Bio-Rad #5000113, #5000114, #500011). 600 μg aliquots of each sample were pre-heated at 95 °C for 5 min. and then loaded on a 6% polyacrylamide gel. After SDS-PAGE migration and

western blot transfer to Immobilon-P membrane (Millipore IPVH00010), the membrane was incubated with 5X blocking solution (0.75 g powder milk in 15 mL 1X PBS, 0.1% Tween20) for 30 min. and then overnight with the primary antibody at 4 °C using antibody-specific dilutions (anti-FLAG-HRP, 1:15000; anti-HA-HRP, 1:15000; anti-GFP, 1:20000; or Anti-NRPD2, 1:2500). Next, the membrane was washed with 15 mL of 1X PBS, 0.1% Tween20 for 20 min., three times at room temperature and incubated with a secondary antibody coupled to horseradish peroxidase (1:15000) for 2 h at 4 °C, followed by che-miluminescent detection on a Fusion FX7 Edge (Vilber) using Lumi-Light Plus Substrate (Roche). The membrane was then stripped with Restore PLUS Western Blot Stripping Buffer (Thermo Scientific), washed, blocked and incubated with the next antibody.

## Affinity purification of Pol IV complexes

**NRPD1-3xF, 3xHA-CLSY1, SHH1-3xHA and RDR2-GFP**. Arabi-dopsis flowers were ground in liquid nitrogen and 1.5 g of this fro-zen powder was then mixed with 3 mL of lysis buffer (LB; Supplementary Table 2) at 4 °C for 20 min. The resulting extracts were centrifugated twice at 16,000 × g at 4 °C for 15 min. to obtain clear supernatants. For NRPD1-3xF and 3xHA-CLSY1 IP-MS, the same pro-tocol was used. Each supernatant was incubated with 50 μL of either anti-FLAG or anti-HA antibodies associated with magnetic beads from the corresponding Miltenyi Biotec Isolation kit on a wheel at 10 rpm at 4 °C for 45 min. μMACS DYKDDDK Isolation Kits were used for anti-FLAG IPs and HA Isolation Kits for anti-HA immunoprecipitation (Mil-tenyi Biotec #130-101-591 and #130-091-122). Next, each μColumn (Miltenyi Biotec #130-042-701) was conditioned with 400 μL of lysis buffer. Magnetic bead-incubated samples were loaded in 400 μL bat-ches on the Miltenyi stand. These columns were washed with 4 ×400 μL of wash buffer (WB; Supplementary Table 2), followed by 200 μL of μMACS kit wash buffer. The protein complexes were then eluted into 1.5 mL Eppendorf tubes using 3 ×35 μL of pre-heated 95 °C μMACS kit elution buffer. The samples were heated to 95 °C for 5 min. and stored at −20 °C prior to MS analysis at the Strasbourg-Esplanade Proteomics Facility. For the co-IP experiments shown in Fig. 2f, g, h, the same IP protocol was followed, except that in addition to the above-mentioned kits, the Miltenyi anti-GFP Isolation Kit was used for anti-GFP IPs (Miltenyi Biotec #130-091-125), and eluted protein samples were analyzed by western blot analysis, as described in Ferrafiat et al.[41].

**N- and C-terminal CLSY 3xF IPs**. The affinity purification was per-formed as described in Law et al.[10] using 10 g of Arabidopsis flowers with the following modifications. The IP buffer was supplemented with 10 mM bortezomib (IB, Supplementary Table 2), anti-Flag M2 magnetic beads (Sigma #M8823) were used, and after five bead washes with 1000 μl of fresh IP buffer, the proteins were eluted twice (15 min. at RT) by incubation with 3x-FLAG peptide (Sigma #F4799) at a concentration of 100 mg/mL in PBS buffer. The eluted samples were precipitated using the trichloroacetic acid (TCA) method and heat-dried pellets were stored at 4 °C until MS analysis.

## Liquid chromatography-tandem mass spectrometry proteomics for 3xF-tagged NRPD1 variants and 3xHA-tagged CLSY1

After Pol IV complex immunoprecipitation, the frozen protein samples were transferred to the Strasbourg-Esplanade Proteomics Facility at the Institut de Biologie Moléculaire et Cellulaire (IBMC). Proteins were precipitated by adding 5 volumes of glacial 0.1 M ammonium acetate in 100% methanol, stored for 12 h at −20 °C, washed with 0.1 M ammo-nium acetate in 80% methanol and dried under vacuum. The dry pel-lets were resuspended in 50 mM ammonium bicarbonate, reduced with 5 mM dithiothreitol (DTT) for 10 min. at 95 °C and alkylated with 10 mM iodoacetamide for 30 min. at room temperature in the dark. The resulting proteins with carbamidomethyl groups on their cysteines were digested with 300 ng of sequencing-grade porcine

trypsin (Promega) and injected on an Easy-nanoLC-1000 system coupled to a Q-Exactive+ mass spectrometer (Thermo Scientific) using a data-dependent acquisition strategy with 160 min. gradients. Arabidopsis proteins were identified via comparison to the TAIR10 database (27,222 protein sequences, https://www.arabidopsis.org/), analyzed using the Mascot algorithm (version 2.6.2, Matrix Science) and spectral counts were validated in Proline software (v2.0, ProFI) with analysis parameters that accommodate a false discovery rate of <1% at the peptide spectrum matches and protein levels. Once Arabidopsis protein IDs were determined, statistic comparisons of the spectral counts in different samples were carried out to obtain $p$-values based on the negative binomial distribution from the R package IPinquiry4[77] and adjusted by the Benjamini-Hochberg method. The raw data for these IP-MS analyses are available via ProteomeXchange with the identifier PXD047743.

## Proteomic characterization of affinity purified 3xF-tagged CLSY complexes

For all four batches of IP-MS experiments (see Fig. 3b and Supplementary Data 3), acetone precipitated protein pellets obtained after immunoprecipitation of the CLSY proteins were resuspended in 4 M urea, 100 mM Tris-Cl, pH 8.5. This was followed by reduction and alkylation by the sequential addition of 5 mM tris(2-carboxyethyl) phosphine and 10 mM iodoacetamide. Reduced and alkylated samples were diluted to reduce urea concentration to 2 M followed by proteolytic digestion with Lys-C and trypsin at 37 °C overnight. The digested peptides were subjected to offline SP3-based peptide cleanup[78] and subsequently analyzed by LC-MS/MS. Briefly, peptides were separated by reversed phase chromatography using 75 μm inner diameter fritted fused silica capillary column packed in-house to a length of 25 cm with bulk 1.9 mM ReproSil-Pur beads with 120 Å pores[79]. For batches 1 and 2, the samples were analyzed on a Thermo Fusion Lumos mass spectrometer coupled to a Dionex Ultimate 3000 UHPLC using a data-dependent acquisition strategy with an MS1 resolution (r) of 120 K followed by sequential MS2 scans at a resolution of 15 K[79]. For batch 3, the samples were analyzed using a Thermo Q-Exactive coupled to an easyLC 1000 using a data-dependent acquisition strategy as previously described[80]. For batch 4, the samples were analyzed on a Thermo Fusion Lumos mass spectrometer coupled to a Dionex Ultimate 3000 UHPLC using a data-independent acquisition strategy in which 90 variable isolation windows (resolution = 15K) were employed to collect MS/MS spectra across a 400–1600 m/z range (vDIA)[81]. For batches 1–3, the data generated by LC-MS/MS were analyzed using the MaxQuant bioinformatic pipeline[82]. The Andromeda integrated into MaxQuant was employed as the peptide search engine and the data were searched against the Arabidopsis database (Uniprot Reference UP000006548). Briefly, a maximum of two missed cleavages was allowed. The maximum false discovery rate for peptide and protein was specified as 1%. Label-free quantification (LFQ) was enabled with an LFQ minimum ratio count of 1. The parent and peptide ion search tolerances were set as 20 and 4.5 ppm, respectively. For batch 4, the vDIA data was analyzed using DIA-Umpire to generate pseudo-MS2 spectra from DIA data followed by MSFragger for database searching[83,84]. The MaxQuant and MSFragger output files were subsequently processed for statistical analysis of differentially enriched proteins using the R package IPinquiry4[77]. The raw data for all four batches of IP-MS data are available through the MassIVE repository via the identifier MSV000093500.

## Mass spectrometry data visualization

The data visualization was performed in R (v4.2.3) using the ggplot2[85] and ggpubr[86] libraries. Results from the IPinquiry analysis were loaded and volcano plots were made based on Log$_2$(fold change) and adjusted $p$-value comparisons between samples, highlighting the CLSY proteins,

SHH1, RDR2 and core Pol IV subunits. The same dataset was used to produce balloon plots showing the spectral count and adjusted $p$-values, respectively, as size and color factors.

## CLSY3-NRPD1 co-IPs

For each genotype, flower tissue (stage 12 and younger) was collected, flash-frozen and ground to a fine powder in liquid nitrogen. For each IP, 0.15 or 0.20 g of powder per genotype was resuspended in two volumes of IP buffer (IB; Supplementary Table 2), spun twice (max speed for 10 min. at 4 °C), and the supernatant was added to a new tube with 75 μL of IP buffer-washed magnetic Streptavidin beads (M-280 Dynabeads #11205D). The sample and bead solutions were rotated (30 min. at 4 °C) and unbound proteins were removed by five washes in 1000 μL of fresh IP buffer (2 min. at 4 °C). The proteins were eluted by heating the beads (95 °C for 5 min.) in 70 μL of BME 4X Laemmli buffer (Biorad #1610747). Prior to loading on a 7.5% TGX SDS-PAGE gel (Biorad #5671024), 30 μL of each sample was heated (5 min. at 95 °C) and spun down (max speed for 3 min. at RT). The proteins were separated at RT in running buffer (RB; Supplementary Table 2) for 30 min. at 60 V followed by 1 h 30 min. at 150 V. Proteins were transferred onto a 0.45 μm PVDF Amersham Hybond membrane (#10600023) at 4 °C in transfer buffer (TB; Supplementary Table 2) for 1 h 30 min. at 160 mA. The membrane was blocked in TBS-T containing 3% BSA for 1 h 30 min. at RT and incubated overnight with a monoclonal Anti-FLAG antibody coupled to horseradish peroxidase (Sigma #A8592; dilution 1:10,000 in TBS-T containing 3% BSA). After rinsing the membrane five times for 5 min. in wash buffer (WWB; Supplementary Table 2) at RT under slow agitation, it was incubated with Pierce ECL2 Western Blotting substrate (#PI80196X3) for 5 min. in the dark. The Azur Sapphire system was used for the chemiluminescence imaging.

## RNA extraction and RT-qPCR

Total RNA from Arabidopsis flower tissue was extracted with TRIzol reagent, treated with DNase I (Thermo Scientific #EN0521) and purified using phenol-chloroform and ethanol precipitation. Reverse transcription was performed using SuperScript IV Reverse Transcriptase (Invitrogen #18090050), RiboLock RNase Inhibitor (Thermo Scientific #EO0381) and random hexamer primers (Thermo Scientific #N8080127) on 1 μg of RNA previously treated with DNase I. qPCR was done on the LightCycler 480 II (Roche) using Takyon No ROX SYBR 2X (Eurogentec #UF-NSMT-B0701), the cDNA and specific primers to detect *AtSN1, ONSEN,* or *UBQ10* transcripts (Supplementary Table 1).

## Small RNA blot

Small RNA isolation from Arabidopsis flowers was performed as described in Böhrer et al.[87]. In short, total RNA was extracted using TRIzol and then size-fractionated using the RNA clean-up protocol of an RNeasy Midi kit (Qiagen #75144). Resulting low molecular weight RNA samples were dehydrated using a SpeedVac (SPD111V, Thermo Scientific), then loaded onto a 16% polyacrylamide gel and run for 1 h at 15 W. The RNA was transferred to Hybond-N+ nylon membrane (GE Healthcare #RPN203B) at 300 mA for 2 h at 4 °C and UV-crosslinked. The membrane was washed with 50 mL 2X SSC (Saline sodium citrate), prehybridized for 3 h in 20 mL of PerfectHyb Plus hybridization buffer (Sigma-Aldrich #H7033-1L), and then hybridized overnight at 35-40 °C either with a polynucleotide kinase (PNK) $^{32}$P end-labeled DNA oligo probe or with a Klenow $^{32}$P internally labeled DNA probe mixture. Protocols for PNK and Klenow probe preparation were described in Böhrer et al.[87]. The membrane was washed three times with 20 mL of 0.5% SDS, 2X SSC and then exposed to a phosphor storage screen for seven days. The signal on the screen was scanned using a Typhoon biomolecular imager (Amersham, GE Healthcare). To rehybridize the membrane, the probe was stripped in hot 0.1% SDS solution (85-95 °C) twice for 20 min., and the membrane was then

prehybridized in 20 mL of PerfectHyb Plus hybridization buffer before adding the new probe.

## Small RNA data processing

All samples included in Supplementary Data 5 were processed as follows. Raw reads were trimmed using cutadapt (v1.18)[88] to remove adapters (-a AACTGTAGGCACCATCAAT) and reads shorter than 15 nt (-m 15). Trimmed reads were mapped to the Arabidopsis TAIR10 genome using ShortStack (v3.8.5)[89] allowing one mismatch (--mismatches 1) and allowing multi-mapped reads that will be guided to a specific location with the fractional-seeded algorithm (-mmap f). To meet the characteristics of Pol IV transcription[31], the mapped reads were further filtered to keep only perfectly-mapped reads or reads with a single mismatch at the 3′-ends using the filter function of bamtools (v2.5.1)[90] together with a previously published script JSON_findPerfectMatches_and_TerminalMisMatches_v3[11]. To facilitate small RNA quantification, Tag Directories were generated from the filtered bam files using the makeTagDirectory function of HOMER (v4.10)[91] with the following options: -format sam -mis 1 -keepAll. Split Tag Directories with a certain smRNA size (21-24 nt) were made using a previously published perl script splitTagDirectoryByLength.dev2.pl[11].

## Core small RNA cluster comparison

To validate the use of the previously reported master set of small RNA clusters ($n = 12,939$; Zhou et al.[21]), we calculated how many 24 nt siRNAs from the current dataset and the Ferrafiat et al.[41] dataset were covered by these clusters. To count the number of 24 nt siRNAs covered by the master set of small RNA clusters, we first used BBmap to filter the bam files to retain reads that are 24 nt in length (reformat.sh minlength = 24 maxlength = 24) and then keep reads that are located within specific regions using samtools[92]. The pie charts showing the 24 nt siRNAs covered by each of the categories were plotted in R.

## Differential expression analysis

To identify differentially expressed 24 nt siRNA clusters compared to the WT controls, DESeq2[93] was used to perform the differential analysis. As 24 nt siRNAs in some of the RdDM mutants (i.e., *pol iv* mutants) are dramatically decreased, siRNAs of this size class cannot be used to estimate the library size. Instead, we used all reads that are mapped to the TAIR10 genome and also meet the criteria of the filtering process (see the Small RNA data processing section) to calculate the library size factors. Differentially expressed siRNA clusters were identified with fold change (FC) $\geq 2$, which is equivalent to $|\log_2 FC| \geq 1$, and a false discovery rate (FDR) < 0.01.

## Visualization of 24 nt siRNAs

Normalized 24 nt siRNAs at the master set of small RNA clusters was generated using the annotatePeaks.pl function of HOMER[91] with the "-size given -fpkm -len 1" options. Violin plots comparing 24 nt siRNA levels across different sets of clusters were made using the ggplot2[85] package of R. Half violin plots and volcano plots were made using the ggplot2[85] package of R based on the DESeq results.

## DNA methylation detection

Genomic DNA extraction was conducted using the Nucleon Phytopure Kit (Cytiva #RPN8511), treated with RNase A/T1 (2 mg/mL) and purified using phenol-chloroform and ethanol precipitation. 500 ng of DNA were digested with the methylation-sensitive HaeIII restriction enzyme and qPCR was conducted to amplify undigested DNA with primers flanking the HaeIII sites in the *AtSN1* retroelement (Supplementary Table 1).

## Heat stress and retrotransposon detection

Seeds were sterilized with 70% ethanol and 4% bleach and grown on solid 0.5X MS medium (Murashige & Skoog, M0255, Duchefa) (1% sucrose, agar, pH 5.7) under long-day conditions (16 h light at 21 °C; 8 h dark at 18 °C). 14-day-old plants were incubated in liquid MS medium under control stress (24 h at 21 °C) or heat stress (24 h at 37 °C). RNA was extracted following the protocol in Böhrer et al.[87], the RNA was DNase I-treated then purified, cDNA was made using random hexamers, and qPCR was executed with transcript-specific primers (Supplementary Table 1) to measure *ONSEN* transcript levels, as described in Ito et al.[16] and Thieme et al.[60].

## Reporting summary

Further information on research design is available in the Nature Portfolio Reporting Summary linked to this article.

## Data availability

The source data for Figs. 2, 4, and Supplementary Figs. 5, 7, are provided as a Source Data file. The IP-MS datasets obtained in this work are available at the PRIDE server via the identifier PXD047743, and at the MassIVE server via identifier MSV000093500. The Illumina smRNA-seq datasets collected for the study were deposited in the NCBI Gene Expression Omnibus (GEO) and are available via the accession numbers GSE242191 and GSE278181. Previously published smRNA-seq datasets, reanalyzed here, are described in Supplementary Data 5. All other data supporting this study are available from the corresponding authors upon request. Source data are provided with this paper.

## Code availability

The bioinformatics code used for processing, analyzing and visualizing the results is available at https://github.com/toddblev/Felgines_Rymen_Martins_2024 and the version used was also deposited[94] at the Zenodo database (https://doi.org/10.5281/zenodo.13890517).

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

## Acknowledgements

We thank lab members and colleagues for helpful comments and fruitful discussions. The Q-Exactive+ mass-spec instrumentation was funded by the IdEx Equipement mi-lourd grant (University of Strasbourg 2015). J.W. was supported by the NIH (GM089778). F.W. was supported by the Swiss National Science Foundation with a Swiss Postdoc Fellowship (project 210561). J.L. was supported by the NIH (GM112966). G.X. was supported by a postdoctoral fellowship from the Paul F. Glenn Center for Biology of Aging Research at the Salk Institute. L.M. was supported by the Catharina Foundation postdoctoral fellowship. This work was also supported by the NGS Core Facility of the Salk Institute with funding from NIH-NCI CCSG: P30 CA01495, NIH-NlA San Diego Nathan Shock Center P30 AG068635, the Chapman Foundation and the Helmsley Charitable Trust. T.B. was supported by the ANR (ANR-17-CE20-0004), and by the Interdisciplinary Thematic Institute IMCBio+ (ITI 2021-2028 program), including funds from IdEx University of Strasbourg (ANR-10-IDEX-0002), SFRI-STRAT'US (ANR-20-SFRI-0012) and EUR IMCBio (ANR-17-EURE-0023) within the framework of the France 2030 program.

## Author contributions

Molecular cloning and transgenic lines were generated by B.R., L.F., M.Z., L.M., J.E., C.C. and M.B. Protein extractions and IPs were performed by L.F., B.R., L.M., P.H., C.H., M.Z. and J.E. Proteomics analyses were conducted by V.P., J.C., L.K., J.W., P.H. Structural modeling was done by F.W. Bioinformatics analyses were performed by G.X., C.M., L.F., B.R., L.M., C.C. and T.B. The manuscript was written by B.R., L.F., L.M., J.L. and T.B.

## Competing interests

The authors declare no competing interests.
