## [Transparent Peer Review file · Nature Communications]

CLSY docking to Pol IV requires a conserved domain critical for small RNA biogenesis and transposon silencing.

Corresponding Author: Dr Todd Blevins

Version 0:

Reviewer comments:

Reviewer #1

(Remarks to the Author)

This is a nice work to explore the biochemical basis of regulation of Pol IV by CLSYs. The author represents a very classic research procedure to identify the functions of key regions. By evolution comparison of multiple sequences, some key sites on Pol IV NRPD1 were identified. The mass-spectrum-based analysis and structural modeling by Alphafold revealed that some conserved regions of Pol IV are responsible for binding to CLSYs. Further, in vivo data confirmed that these specific motif-mediated Pol IV-CLSY interactions are crucial for their function. Generally, the Alphafold-based modeling is a good supplementary to the experimental structure and the validation strengthens the prediction, this study represents an appropriate use of this new tool. I have only some minor suggestions below.

In the abstract, Line 37, change “phenocopy pol iv null mutants” to “phenocopy pol iv null and clsy quad mutants” seems more appropriate. The CLSY docking site mutation on Pol IV does not affect the Pol IV activity but affects the regulation of Pol IV by CLSYs. Although pol iv null mutant resembles the clsy quad mutant, the functional role of CLSY should also be emphasized in this comparison.

Line 82, the measuring mechanism of 30-nt by Pol IV-RDR2 was recently proposed by Xie et al. Trends Biochem Sci. 2024;49(3):247-256, which should be properly cited.

Line190, it is the N-terminal but not the C-terminal.

Reviewer #2

(Remarks to the Author)

Reviewer #3

(Remarks to the Author)

Pol IV is required for siRNA generation and de novo DNA methylation of TEs. Previous studies by Julie Law group have shown that the targeting of Pol IV to chromatin requires the chromatin remodeling factors CLSY1-4. In the submitted manuscript the authors identified the conserved CYC-YPMF motif of Pol IV, which is necessary for binding of Pol IV to CLSY1-4 and siRNA production, DNA methylation, and TE silencing. Overall, their results are convincing and of high technical quality. Although I feel that novel biological insight into this pathway may be somewhat limited, this paper should certainly be published. Specific comments and suggestions are listed below.

1. The CLSY-docking site in NRPB1 is conserved across the plant species. Conservation or evolution of CLSY proteins across species could also be characterized as in Fig 1c. The evolution of CLSY members could be interesting as they are evolving with overlapping functions coping with TEs of diverse genomic localization and sequences. Although extensive analysis of CLSY evolution may be out of scope of this paper, the conservation of CLSY, including non-vascular plants,

should at least be discussed, as the CLSY-docking site was identified by assuming the conserved interaction across the plant species.

2. In Supplementary Figure 5b, they performed the western blotting analyses after immunoprecipitation with FLAG-tagged NRPD1 and its mutation within CYC-YPMF motifs. They show that other subunit of RNA polymerase IV, NRPDE2, was co-purified with NRPD1 regardless of its mutation within CYC-YPMF, which is essential for the co-purification with CLSYs. However, they did not explain why they show two lanes for IP, are they biological replicates? A puzzling thing for me is that no signals are detectable for input and flow through samples even for the transgenic lines. The signals are seen in Supplementary Figure 5c. It is confusing and would be better to explain. In addition, it would be desirable to show other blots (i.e. histone proteins) to clarify that the authors used same amount of protein extract for the Col-0 WT.

3. It is interesting that the nrpd1-49 (G72E) mutation affects siRNA production less for the CLSY3,4 targets. I wonder how that G72 is conserved across the plant species. Possible variations there might also be related to the evolution of CLSY members (point 1 above).

4. (line 389-391) "After 24 h of 37°C treatment ONSEN accumulates to a much higher level in nrpd1-3 null plants compared to WT (Col-0WT) controls (Fig. 4E, orange panel). Does that mean ONSEN transcript?"

Reviewer #4

(Remarks to the Author)

Version 1:

Reviewer comments:

Reviewer #3

(Remarks to the Author)

I do not have any further suggestions to improve the manuscript.

Reviewer #4

(Remarks to the Author)

The reviewer comments are in black text. Our responses are in blue text.

Reviewer #1:

This is a nice work to explore the biochemical basis of regulation of Pol IV by CLSYs. The author represents a very classic research procedure to identify the functions of key regions. By evolution comparison of multiple sequences, some key sites on Pol IV NRPD1 were identified. The mass-spectrum-based analysis and structural modeling by AlphaFold revealed that some conserved regions of Pol IV are responsible for binding to CLSYs. Further, in vivo data confirmed that these specific motif-mediated Pol IV-CLSY interactions are crucial for their function. Generally, the AlphaFold-based modeling is a good supplementary to the experimental structure and the validation strengthens the prediction, this study represents an appropriate use of this new tool. I have only some minor suggestions below.

We thank the reviewer for recognizing the quality and importance of our study. Indeed, AlphaFold2 predictions provided highly complementary information to gain structural insights and help us choose targets for site-directed mutagenesis and downstream molecular analyses.

In the abstract, Line 37, change “phenocopy pol iv null mutants” to “phenocopy pol iv null and clsy quad mutants” seems more appropriate. The CLSY docking site mutation on Pol IV does not affect the Pol IV activity but affects the regulation of Pol IV by CLSYs. Although pol iv null mutant resembles the clsy quad mutant, the functional role of CLSY should also be emphasized in this comparison.

Thank you for this suggestion. We agree and have edited the abstract accordingly.

Line 82, the measuring mechanism of 30-nt by Pol IV-RDR2 was recently proposed by Xie et al. *Trends Biochem Sci.* 2024;49(3):247-256, which should be properly cited.

We agree that this review article is highly relevant to cite here, as suggested by the reviewer. We have modified our manuscript to include this citation and proposed mechanism, lines 81-84: “Pol IV transcribes DNA into short transcripts^{12,30,31} with lengths measured by structural aspects of the Pol IV-RDR2 complex and transcription bubble^{29,32}, and which serve as templates for RDR2 to synthesize a second RNA strand.”²⁹

And added to the References:

²⁹Singh, J., et al. Reaction Mechanisms of Pol IV, RDR2, and DCL3 Drive RNA Channeling in the siRNA-Directed DNA Methylation Pathway. *Mol Cell* 75(3), 576-589 (2019).

³²Xie, G., Du, X., Hu, H. & Du, J. Molecular mechanisms of the RNA polymerases in plant RNA-directed DNA methylation. *Trends Biochem. Sci.* 49, 247–256 (2024).

Line190, it is the N-terminal but not the C-terminal.

Indeed, “N-terminal” is what we intended. Thank you for catching this typo (now Line 192).

Reviewer #2:

We thank this early career researcher for helping to peer review our manuscript.

Reviewer #3:

Pol IV is required for siRNA generation and de novo DNA methylation of TEs. Previous studies by Julie Law group have shown that the targeting of Pol IV to chromatin requires the chromatin remodeling factors CLSY1-4. In the submitted manuscript the authors identified the conserved CYC-YPMF motif of Pol IV, which is necessary for binding of Pol IV to CLSY1-4 and siRNA production, DNA methylation, and TE silencing. Overall, their results are convincing and of high technical quality. Although I feel that novel biological insight into this pathway may be somewhat limited, this paper should certainly be published. Specific comments and suggestions are listed below.

We thank the reviewer for these supportive words, concerning the quality of our results. In addition to providing direct evidence for the “one CLSY per Pol IV complex” model (Fig. 3), and identifying the CLSY docking site on the NRPD1 subunit (Fig. 1 and Fig. 2) that is essential for siRNA production throughout the genome (Fig. 4), our manuscript also suggests a novel mechanism for regulating DNA methylation patterns in plants that includes competition between the CLSYs for binding to the Pol IV complex. In this model, the expression patterns of the CLSYs, their preferences for associating with Pol IV, and their targeting to different chromatin states all contribute to the generation of distinct DNA methylation patterns, moving the field closer to understanding how DNA methylation is modulated during plant development.

1. The CLSY-docking site in NRPB1 is conserved across the plant species. Conservation or evolution of CLSY proteins across species could also be characterized as in Fig 1c. The evolution of CLSY members could be interesting as they are evolving with overlapping functions coping with TEs of diverse genomic localization and sequences. Although extensive analysis of CLSY evolution may be out of scope of this paper, the conservation of CLSY, including non-vascular plants, should at least be discussed, as the CLSY-docking site was identified by assuming the conserved interaction across the plant species.

Thank you for this suggestion. Indeed, an extensive analysis of CLSY evolution is currently underway, but this is outside the scope of the current manuscript. However, we agree that CLSY protein evolution is a point deserving attention in our discussion section, because it's a promising avenue for future research concerning the docking of Pol IV to the distinct SHH1-CLSY1, CLSY3 and CLSY4 recruitment machineries. We have therefore added the following lines and citations to the last paragraph of our revised discussion:

“To further test our model, CLSY motifs that facilitate docking to Pol IV and differentiate their activities from other RNA polymerase recruitment factors could be explored by comparing amino acids in the CLSYs to those of DRD1, an SNF2 domain protein that is required for Pol V transcription. Intriguingly, CLSY functional diversity appears to have increased in flowering plants, which typically have three or more distinct CLSYs, whereas nonvascular plants such as *P. patens* have only a single, CLSY1-like protein.”^{37,68} (Lines 521-526)

And added to the References:

³⁷Trujillo et al. (2018) *Mol Biol Evol.* doi: 10.1093/molbev/msy146.

⁶⁸Chakraborty et al. (2024) *Genome Biol Evol.* doi: 10.1093/gbe/evae119.

2. In Supplementary Figure 5b, they performed the western blotting analyses after immunoprecipitation with FLAG-tagged NRPD1 and its mutation within CYC-YPMF motifs. They show that other subunit of RNA polymerase IV, NRPDE2, was co-purified with NRPD1 regardless of its mutation within CYC-YPMF, which is essential for the co-purification with CLSYs.

Thank you for these comments. We address them point-by-point below.

However, they did not explain why they show two lanes for IP, are they biological replicates?

The western analysis in suppl. Fig. S5B supports the IP-MS balloon plot in primary Fig. 2B, where we show 2 technical replicates each for 3 biologically independent lines per NRPD1-3xF variant, leading to 6 columns of IP-MS balloons per variant. In Fig. S5B, we analyze IP, input and flow through aliquots (without technical replication) from the first two biological replicates of each NRPD1-3xF variant in Fig. 2B. The third biological replicate in Fig. 2B was not included because of the limited number of wells available in the gel used for the western.

To help clarify this point, we have added the following sentence to the Fig. S5B legend: "Because of the limited wells available in the SDS-PAGE gel, this western analysis only includes the first two biological replicates of each NRPD1-3xF variant represented in **Fig. 2B.**" (Lines 1352-1354)

A puzzling thing for me is that no signals are detectable for input and flow through samples even for the transgenic lines. The signals are seen in Supplementary Figure 5c. It is confusing and would be better to explain.

We understand that comparison of the western analyses in Fig. S5B and Fig. S5C could be confusing. However, these IP experiments were performed using different protein extraction protocols, different tagged bait proteins, different beads and different exposure times for the chemiluminescence detection (see separate descriptions in our Methods: "Affinity purification of Pol IV complexes - NRPD1-3xF..." for Fig. S5B and "CLSY3-NRPD1 co-IPs" for Fig. S5C).

In our experience, when detecting Pol IV subunits by western blot following an IP, it is not unusual to have weak or no signals in the inputs and flow throughs, as observed in Fig. S5B. Using Miltenyi Biotec columns with μ MACS Anti-FLAG beads, the strong Pol IV enrichment in the NRPD1-3xF IPs led us to select a short chemiluminescence exposure of 20 s (Fig. S5B). By contrast, to achieve suitable protein detections in the Streptavidin IP experiments (Fig. S5C), exposure times of 5 min. and 30 min. were required. Thus, signals for CLSY3-3xF-BLRP and NRPD1-3xF proteins are detected in input and unbound fractions in the latter western results.

To help clarify this point, we have added labels indicating the short chemiluminescence exposure times " α FLAG (NRPD1); 20 s" and " α NRPDE2; 20 s" in revised Fig. S5B.

In addition, it would be desirable to show other blots (i.e. histone proteins) to clarify that the authors used same amount of protein extract for the Col-0 WT.

The Fig. S5B western analysis was conducted over three years ago, so we do not have these particular protein samples with which to perform a western detection of histone proteins, as requested. However, this is a robust protocol using equal masses of input flowers and equal volumes of liquid nitrogen-ground tissue prior to protein extraction. We have never had problems with significant deviations in total protein in the case of Col-0_{WT} samples.

3. It is interesting that the *nRPD1-49* (G72E) mutation affects siRNA production less for the CLSY3,4 targets. I wonder how that G72 is conserved across the plant species. Possible variations there might also be related to the evolution of CLSY members (point 1 above).

The *nripd1-49* (G72E) mutation was isolated in Ferrafiat et al. (2019), where we noted that Arabidopsis NRPD1 Glycine-72 is conserved in the largest subunits of Pol II, Pol IV and Pol V, including in yeast Pol II (Ferrafiat et al., Fig. S2A). In our latest phylogenetic analysis, we confirmed that Glycine is conserved at this position in all the RNA polymerases and species analyzed (Table S1). Indeed, this Glycine is also conserved in nonvascular plants, like *P. patens* which only have one CLSY protein (see our response to Point#1), so we do not see a correlation between this Glycine in the clamp domain and the evolution of specific CLSY members.

4. (line 389-391) “After 24 h of 37°C treatment ONSEN accumulates to a much higher level in *nripd1-3* null plants compared to WT (Col-0WT) controls (Fig. 4E, orange panel). Does that mean ONSEN transcript?”

Thank you for suggesting this clarification. We have revised the manuscript to indicate “...*ONSEN* transcripts accumulate to a much higher level...”

Reviewer #4:

We thank this early career researcher for helping to peer review our manuscript.